# A lipophilic cation protects crops against fungal pathogens by multiple modes of action

Gero Steinberg[1,2,3 ✉], Martin Schuster [1,3], Sarah J. Gurr[1,2], Tina A. Schrader [1], Michael Schrader [1], Mark Wood[1], Andy Early [1] & Sreedhar Kilaru [1]

The emerging resistance of crop pathogens to fungicides poses a challenge to food security and compels discovery of new antifungal compounds. Here, we show that mono-alkyl lipophilic cations (MALCs) inhibit oxidative phosphorylation by affecting NADH oxidation in the plant pathogens *Zymoseptoria tritici*, *Ustilago maydis* and *Magnaporthe oryzae*. One of these MALCs, consisting of a dimethylsulfonium moiety and a long alkyl chain ($C_{18}$-$SMe_2^+$), also induces production of reactive oxygen species at the level of respiratory complex I, thus triggering fungal apoptosis. In addition, $C_{18}$-$SMe_2^+$ activates innate plant defense. This multiple activity effectively protects cereals against Septoria tritici blotch and rice blast disease. $C_{18}$-$SMe_2^+$ has low toxicity in *Daphnia magna*, and is not mutagenic or phytotoxic. Thus, MALCs hold potential as effective and non-toxic crop fungicides.

[1] Biosciences, University of Exeter, Stocker Road, Exeter EX4 4QD, UK. [2] University of Utrecht, Padualaan 8, Utrecht 3584 CH, The Netherlands. [3]These authors contributed equally: Gero Steinberg, Martin Schuster. ✉email: G.Steinberg@exeter.ac.uk

Fungi pose the greatest biotic challenge to plant health and thus to food security[1]. Intensive monoculture cropping provides an ideal environment for emergence of fungicide-resistant strains[2]. This is of particular concern as ~85% of currently-used fungicides target single enzymes and can be overcome by single point mutations. Indeed, resistance development against azoles, succinate dehydrogenase inhibitors (SDHIs) and QoI inhibitors (strobilurins) challenges agricultural security[3–5]. An alternative approach is use of multi-site fungicides, which usually interfere with unknown essential cellular processes in multiple ways. However, this broader activity often comes with increased toxicity. This is well illustrated by the recent European Union ban of chlorothalonil, a thiol-reactive fungicide pivotal for disease control, but with reported toxicity to aquatic organisms[6] and bumblebees[7]. We thus face an immediate need for new fungicides with multi-site modes of action (MoA) which are environmentally-benign and of low mammalian toxicity[3,8].

One potential target for fungicide development are fungal mitochondria. These organelles are involved in a range of cellular processes, including ATP production by oxidative phosphorylation[9]. This ATP-generating process depends on electron transfer through mitochondrial respiration chain complexes in the inner mitochondrial membrane (IMM)[10]. Fungal mitochondria differ from their mammalian counterparts in composition and respiratory enzyme inventory[11–13]. This makes them an attractive target for fungicide development[14]; indeed, two of the market leader single-target site fungicides, the SDHIs and strobilurins, disrupt the fungal mitochondrial respiration chain[8] (Supplementary Fig. 1a). Mitochondria also produce reactive oxygen species (mROS) at complex I and III, which, if deregulated, damage proteins and lipids in the IMM[15] and trigger apoptotic cell death[16]. Increasing evidence suggests that this programmed cell death pathway exits in fungi[17]; targeting this is a promising strategy for anti-fungal development[18].

Electron-transfer through the respiration chain enables proton transport across the IMM. This renders the matrix of mitochondria negatively-charged, thus making the organelle a target for lipophilic cations. These amphiphilic molecules combine a cationic head group with a highly lipophilic moiety[19] that confers overall lipophilicity to the molecule, indicated by a positive LogP value. This parameter is a widely-accepted descriptor of lipophilicity[20], and positive values are crucial for passive diffusion through bio-membranes[21]. On arrival in mitochondria, the negatively-charged matrix captures the cationic head groups, while the lipophilic moiety inserts into the IMM[22] (Supplementary Fig. 1b). This concentrates lipophilic cations in the mitochondria, thus enabling therapeutics or dyes to enter the organelle (see Supplementary Table 1 for examples). However, an unwanted side-effect of the insertion into the IMM could be inhibition of respiratory enzymes[22–24]. Whilst such features challenge the use of lipophilic cations in therapeutics, they could potentially be exploited in developing anti-fungals.

Mono-alkyl lipophilic cations (MALCs), also known as "cationic surfactants", show anti-bacterial activity[25] and are effective against human pathogenic fungi[26]. Moreover, the lipophilic cation dodecylguanidinium (named $C_{12}$-$G^+$; Supplementary Fig. 1c) is the active ingredient of Syllit (former name dodine), a fungicide widely used to control apple scab and other fruit and foliar orchard diseases, e.g.,[27]. MALCs are amphipathic and, as such, are likely to insert into the plasma membrane[25]. Indeed, all anti-fungal MALCs known to-date are reported to kill fungal cells by altering plasma membrane permeability or function, or by interacting with the fungal cell wall[26,28,29]. However, early reports on dodine suggested that $C_{12}$-$G^+$ enters the cell and inhibits vital metabolic enzymes[30,31]. This casts doubt on a simple mode of action (MoA) of MALCs in affecting the plasma membrane alone.

Consequently, the FRAC code© list 2019 (https://www.frac.info) lists the MoA of dodine as "unknown (U12)".

Here, we investigate the potential of MALCs to control fungal crop diseases. We predominantly use the Septoria blotch fungus *Zymoseptoria tritici*, which seriously threatens temperate-grown wheat[32], and for which we established a raft of live-cell imaging marker strains e.g.,[33,34]. We show that $C_{12}$-$G^+$ targets fungal mitochondria and strongly inhibits ATP synthesis by reducing NADH oxidation and depolarizing the inner membrane. We confirm this multi-site effect on ATP synthesis in two other important crop pathogens, the corn smut fungus *Ustilago maydis* and the rice blast fungus *Magnaporthe oryzae*. Monitoring this MoA, we tested the anti-fungal activity of commercially available and newly-synthesized MALCs that differ in their alkyl-chain length and head group moiety. This led to identification of $C_{18}$-$SMe_2^+$, which triggers the formation of mROS at complex I, which, in turn, results in apoptotic cell death. This dual MoA, combined with activation of plant defense, enables $C_{18}$-$SMe_2^+$ to efficiently protect rice and wheat against rice blast and Septoria tritici blotch disease. Moreover, $C_{18}$-$SMe_2^+$ is not phytotoxic, has low impact on human cells and *Daphnia magna* and is not genotoxic.

## Results

**The lipophilic cation $C_{12}$-$G^+$ has a minor effect on the plasma membrane**. Fungal growth was inhibited in a concentration dependent manner on $C_{12}$-$G^+$ agar plates (Fig. 1a; 50% inhibition at $EC_{50,Z.\ tritici} = 0.76\ \mu g\ ml^{-1}$). To assess the toxicity of $C_{12}$-$G^+$ on *Z. tritici* cells in liquid medium, we stained treated cells with a LIVE/DEAD™ Fixable Red Dead Cell Stain (ThermoFisher, UK). We used *Z. tritici* strain expressing fluorescent plasma membrane marker GFP-Sso1[33] (see Supplementary Table 2 for genotype of all strains and Supplementary Table 3 for experimental strain use). In this assay, live cells are fluorescent green, whereas dying or dead cells show yellow to bright red colouration upon addition of the membrane-impermeable "live/dead dye" (Fig. 1b). We found that $C_{12}$-$G^+$ was effectively killing *Z. tritici* in liquid culture (>80% of cells after 1 h at 100 μg ml$^{-1}$; Fig. 1c). We thence investigated $C_{12}$-$G^+$ effects after ~30 min treatment, at concentrations up to 100 μg ml$^{-1}$, when most treated cells were still alive. This shorter treatment time and lower dose promised to provide insight into the primary cellular response to $C_{12}$-$G^+$.

$C_{12}$-$G^+$ is thought to act on the fungal plasma membrane[26]. We investigated effects on membrane appearance, using GFP-Sso1-expressing cells. Indeed, high concentrations of $C_{12}$-$G^+$ induced formation of GFP-Sso1 "patches" at the cell periphery (Fig. 1d, e), and electron microscopy studies revealed these as plasma membrane invaginations (Fig. 1f and Supplementary Fig. 2). These infolds could be due to excessive insertion of $C_{12}$-$G^+$ into the membrane. Next, we tested if $C_{12}$-$G^+$ affects plasma membrane integrity. We treated cells with increasing concentrations of $C_{12}$-$G^+$ and added propidium iodide. This dye is slightly larger than ATP ($MW_{PI} = 668.41\ g\ mol^{-1}$; $MW_{ATP} = 507.18\ g\ mol^{-1}$), and thus requires openings of >0.7 nm to enter the cell (https://bionumbers.hms.harvard.edu). Indeed, the number of propidium iodide-stained cells increased with higher concentrations of $C_{12}$-$G^+$, but only reached ~30% of cells at 100 μg ml$^{-1}$ (Fig. 1g, h). We tested if $C_{12}$-$G^+$ causes even smaller membrane openings. The cellular membrane potential is based on gradients of potassium, sodium and chloride ions. Considering their diameter, holes of <0.4 nm should allow ion passage (https://bionumbers.hms.harvard.edu), resulting in membrane depolarization. We tested for such effect of $C_{12}$-$G^+$ by using the voltage-sensitive green-fluorescent probe bis-(1,3-dibutylbarbituric acid) trimethine oxonol, DiBAC$_4$(3)[35]. We excluded dead cells from the

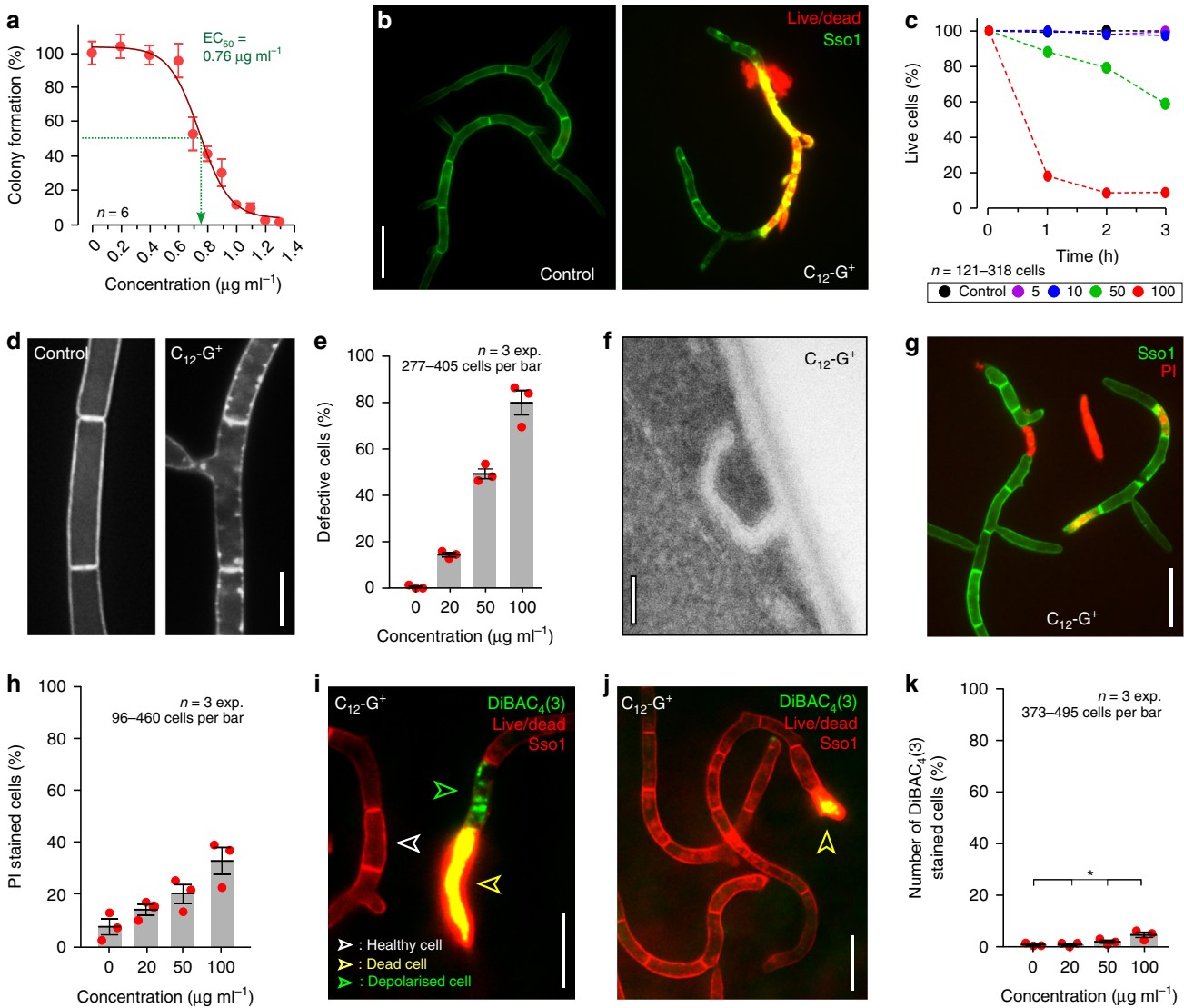

**Fig. 1 The effect of $C_{12}$-$G^+$ on Z. tritici plasma membrane. a** Colony formation of Z. tritici after 5 days growth on agar plates, supplemented with increasing amounts of $C_{12}$-$G^+$. Green dotted line indicates $EC_{50}$ concentration. **b** Live/dead staining of Z. tritici cells, expressing plasma membrane marker GFP-Sso1, after 3 h treatment with $C_{12}$-$G^+$. Dead cells are either yellow or red. Scale bar = 15 μm. **c** Survival curves of Z. tritici cells grown in $C_{12}$-$G^+$-supplemented liquid medium. Scale bar = 5 μm. **d** Plasma membrane, labeled with GFP-Sso1, in $C_{12}$-$G^+$ and solvent-only-treated cells (Control) of Z. tritici. $C_{12}$-$G^+$ induces accumulation of the fluorescent marker at the cell periphery. **e** Bar chart showing the increase of cells with GSP-Sso1 accumulations with increasing concentrations of $C_{12}$-$G^+$. **f** Electron micrographs of plasma membrane invaginations, caused by $C_{12}$-$G^+$ treatment in Z. tritici. See also Supplementary Fig. 2. Scale bar = 0.1 μm. **g** Effect of $C_{12}$-$G^+$ on plasma membrane permeability to the small molecule propidium iodide. Note that propidium iodide has a similar size to ATP. Scale bar = 15 μm. **h** Bar chart showing number of cells that contain propidium iodide at various concentrations of $C_{12}$-$G^+$. **i** Effect of $C_{12}$-$G^+$ on plasma membrane permeability to ions, indicated by membrane depolarization, as monitored with the voltage-sensitive fluorescent dye $DiBAC_4(3)$ (green). Cells express mCherry-Sso1 (red at cell periphery) and were counter-stained with live/dead dye (red). Live cells only show red fluorescence in the plasma membrane (white open arrow), dead cells combine cytoplasmic red and green fluorescence (yellow open arrow); depolarised cells combined a faint peripheral staining in red and cytoplasmic green fluorescence (green open arrowhead). Only these cells were considered in the subsequent quantitative analysis. Scale bar = 10 μm. **j** Z. tritici cells, expressing mCherry-Sso1 (red, peripheral), incubated with $C_{12}$-$G^+$ and co-stained with live/dead stain and $DiBAC_4(3)$. Only dead cells take up the voltage-sensitive dye (yellow open arrowhead), demonstrating that the MALC is not perforating the plasma membrane. Scale bar = 10 μm. **k** Bar chart showing number of cells which are $DiBAC_4(3)$-positive at various concentrations of $C_{12}$-$G^+$. Note that only green-fluorescent cells that did not show live/dead dye uptake are included. Values (**a**, **e**, **h**, **k**) are shown as mean ± standard error of the mean (SEM), sample size *n* is indicated in each panel. Red dots represent data points. Non-linear regression curve (**a**) was calculated as dose-response inhibition (four parameters) in Prism5. In **k**, one-way ANOVA testing was performed; *P value of 0.0107. See Supplementary Table 7 for experimental conditions. All source data are provided as a Source Data file.

analysis by co-staining with a live/dead stain. The living cells showed red-fluorescent plasma membranes, but only took up $DiBAC_4(3)$ upon cell depolarization (Fig. 1i). Even at 100 μg ml$^{-1}$ $C_{12}$-$G^+$, only few cells showed green $DiBAC_4(3)$ fluorescence, suggesting that the MALC has minor effects on the plasma

membrane (Fig. 1j, k). Thus, we conclude that disruption of the plasma membrane is not the primary MoA of $C_{12}$-$G^+$.

**$C_{12}$-$G^+$ alters fungal mitochondrial organization and respiration.** Next, we tested the alternative hypothesis that lipophilic

cation $C_{12}$-$G^+$ targets the negatively-charged mitochondria and interferes with fungal respiration. In a first step, we predicted the Log$P$ value of $C_{12}$-$G^+$, an important factor for passive penetration of the plasma membrane[21], and compared it to other lipophilic cations, known to target mitochondria. This revealed that the lipophilicity of $C_{12}$-$G^+$ is comparable to the mitochondrial dyes Rhodamine 123 or tetramethylrhodamine methyl ester (TMRM[36]; Log$P_{C12\text{-}G+}$ = 2.26; Log$P_{R123}$ = 2.15; Log$P_{TMRM}$ = 2.74; Supplementary Table 1). This suggests that $C_{12}$-$G^+$ has the biophysical properties required to enter and accumulate in the mitochondria. We tested for an effect of $C_{12}$-$G^+$ on morphology of fluorescent mitochondria in living *Z. tritici* cells[33] and found low concentrations of $C_{12}$-$G^+$ induced mitochondrial fragmentation (Fig. 2a, b; EC$_{50}$: 4.12 µg ml$^{-1}$; note that all EC$_{50}$ values provided in this paper were corrected for the molecular weight of the counter ions in the various compounds). Electron microscopy revealed that IMM organization was altered, with disorganized and swollen cristae (Fig. 2c shows control; Fig. 2d). Such changes are indicative of insertion of lipophilic cations into the IMM[37]. The respiratory chain protein complexes in this membrane generate a proton gradient, required to synthesize ATP, leaving the matrix negatively charged. We tested if $C_{12}$-$G^+$ interferes with this gradient by staining $C_{12}$-$G^+$-treated cells with the mitochondrial potential dye TMRM. Indeed, TMRM fluorescence was reduced in $C_{12}$-$G^+$ cells, indicating that the proton gradient over the inner membrane was abolished and mitochondria were depolarized (Fig. 2e, f). This effect occurs at doses below 0.5 µg ml$^{-1}$, suggesting that IMM depolarization represents the primary MoA of $C_{12}$-$G^+$ in fungi.

The proton gradient over the IMM underpins generation of ATP[10]. Thus, depolarizing the IMM should inhibit ATP synthesis. We tested this by measuring ATP concentration in *Z. tritici* cell extracts. Indeed, when cells were treated with $C_{12}$-$G^+$ at concentrations where mitochondria were no longer stained with TMRM (5 µg ml$^{-1}$), ATP levels dropped significantly with time (Fig. 2g). This confirms that $C_{12}$-$G^+$ targets mitochondria and impairs oxidative phosphorylation. We next asked if this effect is due to a block in electron transfer through the respiration chain. Fungi have a "branched respiration chain"[11,12], where electrons are provided by NADH oxidation at respiration complex I and fungal-specific alternative NADH dehydrogenases (Fig. 2h). At complex IV, the electrons are neutralized by formation of water from protons and oxygen (Fig. 2h). We tested oxygen consumption in *Z. tritici* cells, using a modified Winkler assay to test for dissolved oxygen in the culture medium. We treated cells with the solvent only (control) or $C_{12}$-$G^+$ for 3 h in sealed containers and determined the oxygen in the supernatant. We found that 57.9% of the oxygen was consumed in control cells (Fig. 2i), whereas incubation with $C_{12}$-$G^+$ reduced O$_2$-levels by only 18% (Fig. 2i). This confirms a defect in oxidative phosphorylation. Finally, we measured NADH oxidation in isolated mitochondria. We prepared the organelles from a strain that expressed the mitochondrial marker Acd1-ZtGFP[33] and from the wildtype. Green Acd1-ZtGFP labeled mitochondria were stained with TMRM, confirming that most mitochondria were still "healthy" (Fig. 2j; yellow in the merged image). Next, we measured the effect of $C_{12}$-$G^+$ on oxidation of externally added NADH in unlabeled mitochondria, using a colorimetric method[38]. In control experiments, we added the complex I inhibitor rotenone[39] and the broad-spectrum NADH oxidase inhibitor diphenyleneiodonium (DPI)[40], so as to block complex I and the alternative NADH hydrogenases. This treatment reduced NADH oxidation by ~80% (Fig. 2k, Roten.+DPI). When $C_{12}$-$G^+$ was added to the mitochondria preparation, NADH oxidation was significantly inhibited (Fig. 2k, $C_{12}$-$G^+$). Thus, we conclude that $C_{12}$-$G^+$ affects oxidative phosphorylation, which most likely involves inhibition of NADH oxidation and depolarization of the IMM.

We conjected that inhibition of mitochondrial respiration may account for the fungal specificity of $C_{12}$-$G^+$. To test this, we treated human skin fibroblasts (C109) with $C_{12}$-$G^+$ and monitored mitochondrial shape and membrane potential changes. $C_{12}$-$G^+$ had little effect on mitochondrial organization at concentrations up to 20 µg ml$^{-1}$, but beyond this concentration caused organelle fragmentation (Fig. 3a, b) and swollen mitochondrial cristae (Fig. 3c, Supplementary Fig. 3a). This suggests that $C_{12}$-$G^+$ inserts into the human mitochondrial inner membrane, albeit at ~6.3-times higher concentrations than in *Z. tritici* (Table 1; compare EC$_{50}$ values for mitochondrial fragmentation). At high concentrations, $C_{12}$-$G^+$ affected the mitochondrial membrane potential in C109 cells (Fig. 3d; Supplementary Fig. 3b). However, these effects in human fibroblasts occurred at ~46-times higher $C_{12}$-$G^+$ concentration than in *Z. tritici* cells (Table 1; compare EC$_{50}$ values for depolarization). Thus, we speculate that the inhibitory effect on mitochondrial respiration most likely underpins the higher toxicity of $C_{12}$-$G^+$ for fungi.

**The identification of "better" MALCs.** Having established quantitative assays for the MoA in mitochondria, we set out to identify more effective MALCs. As the cationic head group and the length of the alkyl chain reportedly determine MALC activity[24], we synthesized variants and investigated their effect on *Z. tritici* mitochondrial fragmentation and depolarization (Fig. 4a; counter ions listed in figure legend). We tested (i) $C_{12}H_{25}$-alkyl cations with various moieties (dodecyltrimethylammonium= $C_{12}$-$NMe_3^+$; dodecyltriethylammonium=$C_{12}$-$NEt_3^+$; dodecyldimethylsulfonium=$C_{12}$-$SMe_2^+$), MALCs (ii) short $C_6H_7$-alkyl chains (hexyltrimethylammonium=$C_6$-$NMe_3^+$, hexyldimethylsulfonium=$C_6$-$SMe_2^+$) and (iii) longer $C_{18}H_{19}$-alkyl chains (octadecyltrimethylammonium=$C_{18}$-$NMe_3^+$; octadecyldimethylsulfonium=$C_{18}$-$SMe_2^+$). We used lipophilic alkyl anions (dodecylphosphate=$C_{12}$-$PO_4^{2-}$, dodecylsulfate=$C_{12}$-$SO_4^-$) and a symmetric lipophilic (dodecane-1,12-diylbis(dimethyl) sulfonium=$C_{12}$-$(SMe_2^+)_2$), as controls as their negatively charge or the positive charge at both ends would preclude targeting to the mitochondrial matrix or membrane insertion, respectively. At low concentrations, only molecules in group (i) and group (iii) induced significantly higher mitochondrial fragmentation than $C_{12}$-$G^+$ (Fig. 4b and Supplementary Fig. 4a; EC$_{50}$ $_{[C12\text{-}NMe3^+]}$: 2.85 µg ml$^{-1}$; EC$_{50}$ $_{[C12\text{-}NEt3^+]}$: 1.88 µg ml$^{-1}$; EC$_{50}$ $_{[C12\text{-}SMe2^+]}$: 4.83 µg ml$^{-1}$; EC$_{50}$ $_{[C18\text{-}NMe3^+]}$: 0.11 µg ml$^{-1}$; EC$_{50}$ $_{[C18\text{-}SMe2^+]}$: 0.10 µg ml$^{-1}$; compare to EC$_{50}$ $_{[C12\text{-}G^+]}$: 4.12 µg ml$^{-1}$) and induced mitochondrial ultrastructural changes (Fig. 4c and Supplementary Fig. 5). This suggests that only cationic lipophilic molecules with long alkyl chains insert into the mitochondrial membrane. However, mitochondrial depolarization assays revealed that only the two $C_{18}$-alkyl chain cations were more potent than $C_{12}$-$G^+$ in affecting mitochondrial respiration (Fig. 4d and Supplementary Fig. 4b; EC$_{50}$ $_{[C12\text{-}G^+]}$: 0.28 µg ml$^{-1}$; EC$_{50}$ $_{[C18\text{-}NMe3^+]}$: 0.11 µg ml$^{-1}$; EC$_{50}$ $_{[C18\text{-}SMe2^+]}$: 0.20 µg ml$^{-1}$). To confirm this, we measured ATP levels in cytoplasmic cell extracts, treated with $C_{18}$-$NMe_3^+$ and $C_{18}$-$SMe_2^+$ at concentrations, where mitochondria were largely depolarized (5 µg ml$^{-1}$, Supplementary Fig. 4b; note that neither $C_{18}$-$NMe_3^+$ nor $C_{18}$-$SMe_2^+$ abolished TMRM staining). This confirmed a significant inhibitory effect of these MALCs on ATP synthesis within 2 h (Fig. 4e). Similar to $C_{12}$-$G^+$, $C_{18}$-$NMe_3^+$ and $C_{18}$-$SMe_2^+$ treatment reduced NADH oxidation in isolated mitochondria (Fig. 4f), supporting the notion that MALCs reduce electron intake into the respiratory chain. DiBAC$_4$(3) staining of cells, pre-treated with ~50-times the EC$_{50}$ of $C_{18}$-$NMe_3^+$ or $C_{18}$-$SMe_2^+$ (7 and 14 µg ml$^{-1}$ compound, respectively, provided no evidence for damaging effects of either MALC on the *Z. tritici* plasma membrane (Fig. 4g; DiBAC$_4$(3) staining of $C_{12}$-$G^+$ at EC$_{50}$ included for

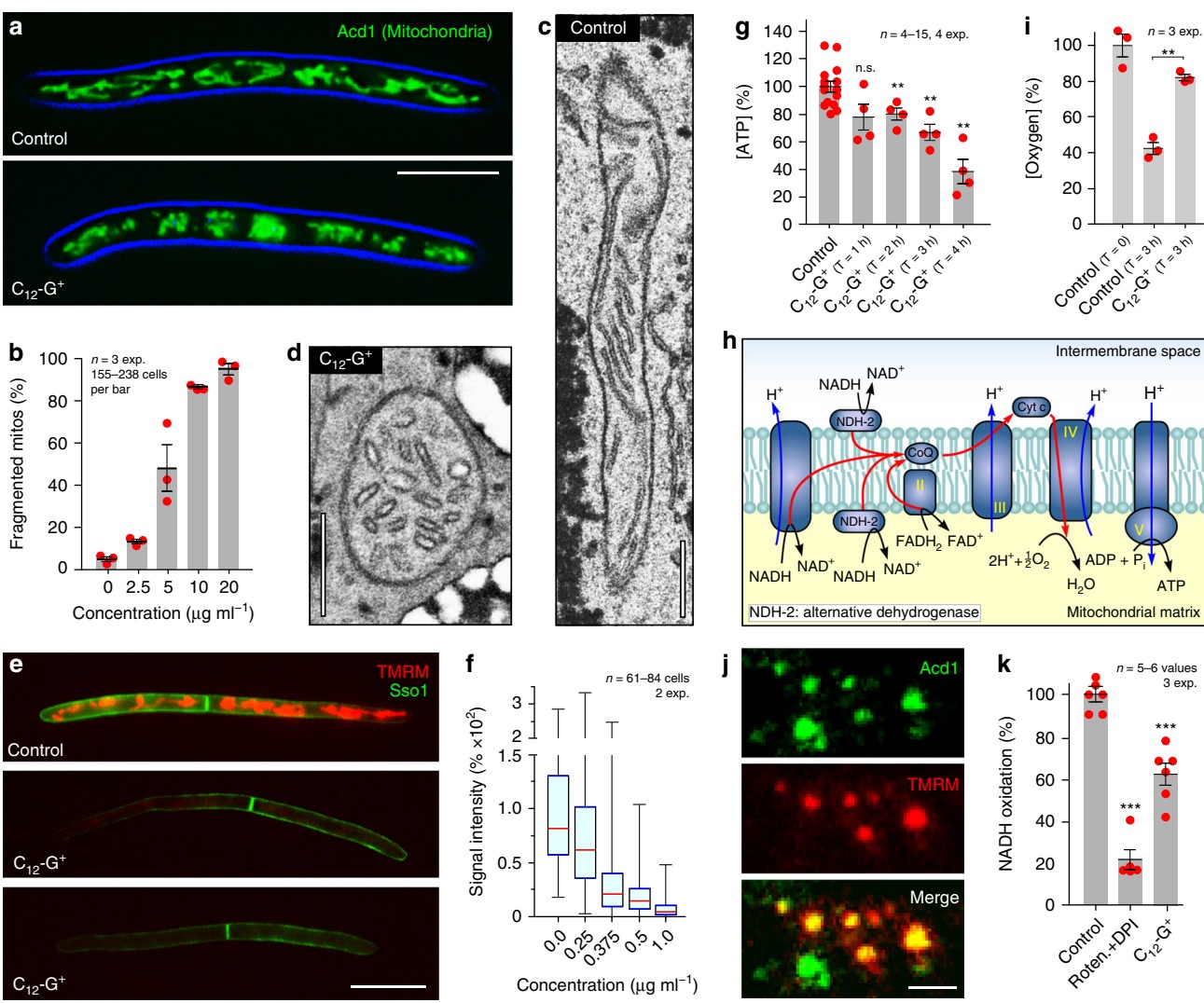

**Fig. 2 The effect of $C_{12}$-$G^+$ on Z. tritici mitochondria. a** Mitochondria in *Z. tritici* in cells treated with solvent methanol (Control) and $C_{12}$-$G^+$. Organelles were stained with fluorescent reporter proteins[33]. Scale bar = 10 μm. **b** Bar chart showing $C_{12}$-$G^+$ effect of mitochondrial fragmentation in *Z. tritici*. **c**, **d** Electron micrographs of mitochondria in methanol-treated *Z. tritici* (Control (**c**), $C_{12}$-$G^+$-treated (**d**)). Note swollen cristae in $C_{12}$-$G^+$-treated cells. Scale bar = 0.3 μm. **e** Mitochondrial membrane potential, visualized with TMRM, in *Z. tritici* cells treated with $C_{12}$-$G^+$. Scale bar = 10 μm. **f** Bar chart showing effect of increasing concentrations of $C_{12}$-$G^+$ on TMRM signal intensity, which reflects the mitochondrial proton gradient in *Z. tritici* cells. **g** Bar chart showing the relative amount of ATP in cell extracts, pre-treated for 1–4 h with $C_{12}$-$G^+$. **h** Schematic overview of the "branched respiration chain" of fungi. Note that NADH oxidation involves complex I, but also fungal-specific alternative NADH dehydrogenases[11,12]. **i** Bar chart showing the consumption of oxygen of *Z. tritici* cells, treated with the solvent methanol (Control) and $C_{12}$-$G^+$. Dissolved oxygen in the medium was measured by Winkler titration. **j** Isolated mitochondria, labeled with the marker GFP-Acd1[33] (green) and stained with the potential dye TMRM (red). Scale bar = 1 μm. **k** NADH oxidation measured in isolated mitochondria, treated with $C_{12}$-$G^+$. Values (**b**, **g**, **i**, **k**) are shown as mean ± SEM. Red dots represent data points. **f** Shows Whiskers' plots, with 25th/75th percentiles (blue lines), medians (red line), and minimum and maximum (whiskers ends). Sample size *n* is indicated in each panel. Student's *t*-testing with Welch correction (**g**, **i**, **k**); n.s. non-significant difference to control $P > 0.05$ (**g**), **$P < 0.032$ to $P < 0.0032$ (**g**), and $P < 0.0018$ (**i**); ***$P < 0.0004$ (**k**). All *P*-values are two-tailed. See Supplementary Table 7 for experimental conditions. All source data are provided as a Source Data file.

comparison). This confirms the notion that MALCs do not disrupt the plasma membrane, but rather inhibit oxidative phosphorylation. Quantitative live/dead staining, after 24 h exposure to 10 μg ml$^{-1}$, demonstrated that $C_{18}$-$NMe_3^+$, $C_{18}$-$SMe_2^+$ are 2–7-times more toxic than $C_{12}$-$G^+$ (Fig. 4h). Thus, strong inhibition of mitochondrial membrane potential and ATP synthesis corresponds with increased fungi-toxicity, again implicating inhibition of respiration as the primary physiological effect of MALCs.

**$C_{18}$-$SMe_2^+$ induces ROS production and programmed cell death.** Defects in fungal complex I activity are reported to increase mROS production in fungi[38]. We tested mROS formation in

*Z. tritici* cells by using the ROS indicator dihydrorhodamine-123 (DHR-123), which, after oxidation by ROS, becomes fluorescent[41]. When control cells were treated with DHR-123, low fluorescence was detected in mitochondria (Fig. 5a, Control; 5b, upper panel). In the presence of rotenone, mROS formation was almost abolished (Fig. 5a, rotenone), which confirms that mROS is produced at complex I. Treatment of *Z. tritici* cells with $C_{12}$-$G^+$ and $C_{18}$-$NMe_3^+$ also significantly decreased mROS (Fig. 5a), which is consistent with the observed inhibitory effect of both MALCs on electron release from NADH. We next tested $C_{18}$-$SMe_2^+$ for its effect on mROS production in *Z. tritici* cells. Surprisingly, we found that $C_{18}$-$SMe_2^+$ acts differently from $C_{12}$-$G^+$ and $C_{18}$-$NMe_3^+$. Exposure to $C_{18}$-$SMe_2^+$ for 30 min and 24 h significantly

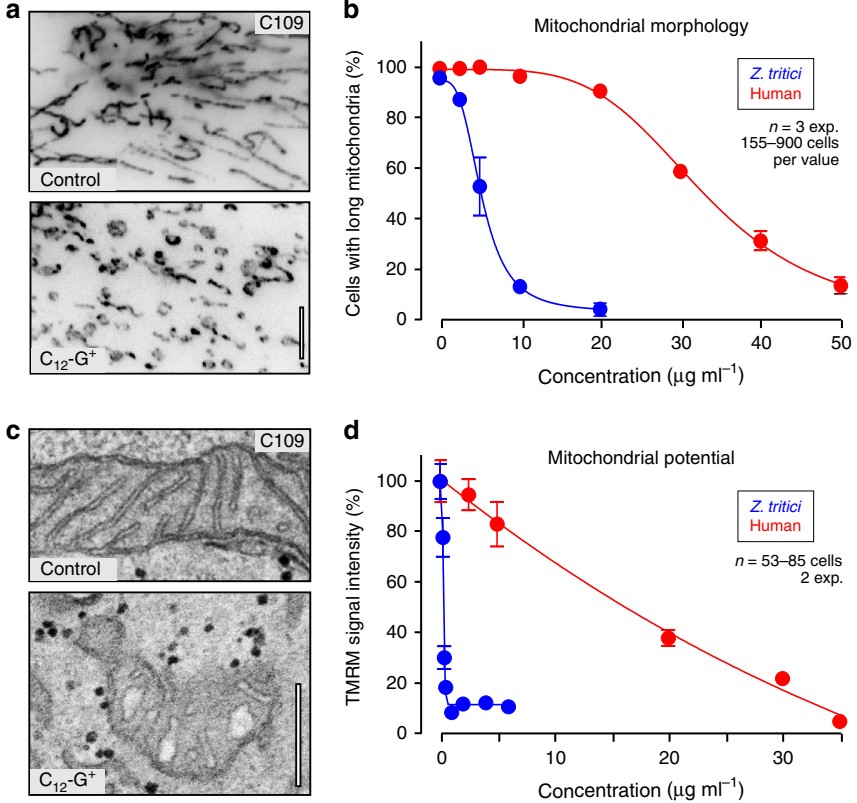

**Fig. 3 Effect of $C_{12}$-$G^+$ on mitochondria in human fibroblasts. a** Mitochondria morphology in human skin fibroblasts (C109 cells), treated with the solvent (Control) and $C_{12}$-$G^+$. Mitochondria were labeled using an antibody against ATP synthase B. Scale bar = 5 μm. **b** Sensitivity of mitochondria morphology to $C_{12}$-$G^+$ in C109 cells (Human) and *Z. tritici* cells. See Table 1 for $EC_{50}$ values. **c** Ultrastructure of mitochondria in human fibroblasts, treated with the solvent (Control) and $C_{12}$-$G^+$. Note swollen cristae in $C_{12}$-$G^+$-treated C109 cells. Scale bar = 0.3 μm. See also Supplementary Fig. 3a. **d** Sensitivity of mitochondrial potential to $C_{12}$-$G^+$ in C109 cells (Human) and *Z. tritici* cells. See Table 1 for $EC_{50}$ values and also Supplementary Fig. 3b. Values (**b**, **d**) are shown as mean ± SEM; sample size *n* is indicated in each panel. Non-linear regression curve was calculated as dose-response inhibition (four parameters) in Prism5. See Supplementary Table 7 for experimental conditions. All source data are provided as a Source Data file.

elevated mROS levels (Fig. 5b, lower panel; Fig. 5c). Co-incubation with rotenone drastically reduced this rise in mROS (Fig. 5c), indicating that $C_{18}$-$SMe_2^+$ induces mROS production at respiration complex I. We next investigated the importance of alkyl chain length in the ability of $C_{18}$-$SMe_2^+$ to trigger mROS formation by DHR-123 staining of cells treated with a newly-synthesized $C_{16}$-alkyl chain dimethylsulfonium cation ($C_{16}$-$SMe_2^+$) and $C_{12}$-$SMe_2^+$ (Supplementary Fig. 6). Surprisingly, 30 min incubation with either MALC did not increase mROS levels, neither at 5 μg $ml^{-1}$, nor at 20 μg $ml^{-1}$ (Fig. 5d). Thus, we conclude that $C_{18}$-$SMe_2^+$ affects complex I, and that the $C_{18}$-alkyl chain is required for this unique activity.

Increasing evidence suggests that fungi can undergo apoptotic cell death[17,18], and mROS induces this pathway[38,42]. Thus, we tested for apoptosis in *Z. tritici* cells, treated with $C_{12}$-$G^+$, $C_{18}$-$NMe_3^+$ and $C_{18}$-$SMe_2^+$. In budding yeast and *Candida albicans*, apoptosis involves metacaspases[38,43,44]. We tested for such caspase activity in $C_{18}$-$SMe_2$-treated *Z. tritici* cells, using the marker CaspACE™ FITC-VAD-FMK, which was been shown to detect apoptotic cell death in fungi[45,46]. We co-stained with propidium iodide to distinguish early apoptotic cells from dead, post-apoptotic cells. After 24 h treatment with all MALCs, only a few apoptotic cells were found after incubation with $C_{12}$-$G^+$ or $C_{18}$-$NMe_3^+$ (Fig. 5e blue arrowheads; Fig. 5f), whereas $C_{18}$-$SMe_2^+$ induced a significant increase in early apoptotic cells (Fig. 5f). We confirmed this result with Annexin-V-fluorescein staining of exposed plasma membrane phosphatidylserine in early apoptotic cells[47] and found membrane-associated fluorescence

only in $C_{18}$-$SMe_2^+$-treated cells (Fig. 5g, h). Thus, we propose that $C_{18}$-$SMe_2^+$-induced mROS production results in activation of a programmed cell death pathway in the pathogen.

**MALCs affect mitochondria in rice blast and corn smut fungi.** Our results show that $C_{12}$-$G^+$, $C_{18}$-$NMe_3^+$, and $C_{18}$-$SMe_2^+$ depolarize the mitochondrial membrane and inhibit ATP synthesis in *Z. tritici*. Moreover, $C_{18}$-$SMe_2^+$ induces mROS production, causing oxidative damage in mitochondria and triggering apoptosis. We investigated if these phenotypes are a general hallmark of this MALC in the crop pathogens *Magnaporthe oryzae* and *Ustilago maydis*. Firstly, we tested if $C_{12}$-$G^+$, $C_{18}$-$NMe_3^+$, and $C_{18}$-$SMe_2^+$ affect appressorium formation in *M. oryzae* germlings, which is a crucial developmental step for rice infection[48]. Within 3 h, conidia treated with the solvent control germinated and formed appressoria (Fig. 6a), but exposure to $C_{12}$-$G^+$, $C_{18}$-$NMe_3^+$, and $C_{18}$-$SMe_2^+$ inhibits germination and appressorium formation (Fig. 6a, b). While the $EC_{50}$ value for all MALCs is very similar (Table 1), $C_{18}$-$SMe_2^+$ and $C_{18}$-$NMe_3^+$ were more effective in inhibiting germination at higher concentrations (Fig. 6b). Next, we tested for an effect of MALCs on *U. maydis*. We found that MALCs efficiently inhibit colony formation on agar plates (Supplementary Fig. 7a; only $C_{12}$-$G^+$ is shown). Consistent with *Z. tritici*, all MALCs altered mitochondrial ultrastructure in both fungi (Supplementary Fig. 7b, c). Significant depolarization of mitochondria accompanied these morphological changes (Fig. 6c, d and Supplementary Fig. 7d shows *U. maydis* cells), confirming that inhibition of oxidative phosphorylation is a general MoA of

**Table 1 Anti-fungal activity and toxicity of MALCs.**

| | $C_{12}$-$G^+$ | $C_{18}$-$NMe_3^+$ | $C_{18}$-$SMe_2^+$ |
|---|---|---|---|
| **Effects in *Z. tritici*** | | | |
| Depolarisation of the plasma membrane[a] | 2.15 | 3.38 | 2.98 |
| Fragmentation of mitochondria[b] | 4.12 | 1.12 | 1.25 |
| Depolarisation of mitochondria[b] | 0.28 | 0.11 | 0.20 |
| Inhibition of ATP synthesis[c] | 19.84 | 54.23 | 58.66 |
| Cell mortality[d] | 32.29 | 89.27 | 66.47 |
| **Effect on human cells** | | | |
| Fragmentation of mitochondria[b] | 26.13 | 35.58 | 35.26 |
| Inhibition of respiration[b] | 12.97 | 9.89 | 9.90 |
| Cytotoxicity (MTT assay, C109)[b] | 2.32 | 4.13 | 3.53 |
| Cytotoxicity (MTT assay, HepG2)[b] | 0.99 | 2.42 | 3.67 |
| Relative mitochondrial toxicity[e] | 54.21 | 88.71 | 47.86 |
| **Toxicity in zooplankton** | | | |
| *Daphnia magna* survival[f,b] | 0.31 | 1.61 | 2.42 |
| Relative toxicity[g] | 10.01 | 143.73 | 160.86 |
| **Phytotoxicity[h]** | | | |
| Leaf symptoms in wheat | >1000 | >1000 | >1000 |
| Leaf symptoms in rice | >1000 | >1000 | >1000 |
| **Anti-fungal protection** | | | |
| Protection against Septoria leaf blotch[i] | 5 | 1.57 | 0.22 |
| Protection against rice blast disease[j] | 17.19 | 12.45 | 2.67 |
| Inhibition of *M. oryzae* germination[b] | 0.83 | 0.77 | 0.76 |
| Induction of plant defence[k] | 5.46 | 0.80 | 14.83 |

[a]Percentage of depolarised cells (DiBAC$_4$(3)-positive) after treatment with ~50-times the EC$_{50}$-value for depolarisation of mitochondria; note one-way ANOVA testing reveals no difference to control (1.95%).
[b]Concentration ($\mu$g ml$^{-1}$) at 50% effect (EC$_{50}$) after 30–45 min treatment. Values were estimated graphically from non-linear regression curves and corrected for the molecular weight of their counter ions; all curves were based on data from 2–3 experiments.
[c]Percent reduction of ATP in cell extracts after 2 h incubation with 5 $\mu$g ml$^{-1}$ MALC; statistically different to control with $P = 0.0146$ ($C_{12}$-$G^+$), $P = 0.0049$ ($C_{18}$-$SMe_2^+$), $P = 0.0068$ ($C_{18}$-$NMe_3^+$).
[d]Percentage of dead *Z. tritici* cells in liquid cultures after 24 h incubation at 10 $\mu$g ml$^{-1}$.
[e]Quotient of EC$_{50}$ values for inhibition of human and fungal respiration.
[f]Number of immobile/non-responsive water fleas at EC$_{50}$ after 24 h.
[g]Mortality in *Z. tritici* multiplied by mortality in *Daphnia*, indicates equivalence in toxicity.
[h]Lowest concentration ($\mu$g ml$^{-1}$) at which chlorosis occurs after 7 days incubation.
[i]Mean area with pycnidia (%; wheat, sprayed with 100 $\mu$g ml$^{-1}$, infected, assessed after 21 days).
[j]Mean lesion area (%; rice treated with 125 $\mu$g ml$^{-1}$, infected and assessed after 4 days).
[k]DAB-stained leaf area after 6 h treatment with 100 $\mu$g ml$^{-1}$, given as percent of positive control.

$C_{12}$-$G^+$, $C_{18}$-$NMe_3^+$, and $C_{18}$-$SMe_2^+$. Finally, we tested mROS production in MALC-treated cells of *M. oryzae* and *U. maydis*. Indeed, only $C_{18}$-$SMe_2^+$ significantly increased mROS levels in both fungi (Fig. 6e, f and Supplementary Fig. 7e shows *U. maydis*). Thus $C_{12}$-$G^+$, $C_{18}$-$NMe_3^+$, and $C_{18}$-$SMe_2^+$ show almost identical effects on mitochondria in all three fungal pathogens.

**$C_{18}$-$SMe_2^+$ protects against wheat blotch and rice blast.** We asked if $C_{18}$-$SMe_2^+$ endows plant protective activity against fungal pathogens. We investigated if $C_{12}$-$G^+$, $C_{18}$-$NMe_3^+$, or $C_{18}$-$SMe_2^+$ adversely affect wheat or rice leaves. We sprayed whole plants to "leaf run off" with 1000 $\mu$g ml$^{-1}$ $C_{12}$-$G^+$, $C_{18}$-$NMe_3^+$, and $C_{18}$-$SMe_2^+$, or water containing small amounts of methanol (=negative control), 10% Tween 20 (=positive control). Despite high concentrations, none of the MALCs induced chlorosis or necrosis in wheat or rice leaves after 7 days (Fig. 7a, b and Supplementary Fig. 8a) and thus $C_{12}$-$G^+$, $C_{18}$-$NMe_3^+$, and $C_{18}$-$SMe_2^+$ are not phytotoxic. Next, we performed quantitative wheat and rice leaf infection assays. We sprayed various concentrations of the three MALCs onto wheat and rice, incubated the plants for 24 h before applying *Z. tritici* (strain IPO323) or *M. oryzae* (strain Guy11), and quantified disease symptoms. In control experiments, *Z. tritici* formed dark "spots" on chlorotic leaves after 21 days, which represent melanised pycnidia, symptomatic of Septoria wheat blotch disease (Fig. 7c and Supplementary Fig. 8b, "Septoria wheat blotch"). Infection with *M. oryzae* resulted in brown disease lesions after 4 days incubation (Fig. 7d and Supplementary Fig. 8b, "Rice blast"). Symptom development was inhibited when leaves were pre-treated with $C_{12}$-$G^+$, $C_{18}$-$NMe_3^+$, or $C_{18}$-$SMe_2^+$ (Fig. 7c, d).

This protective effect was concentration-dependent (Fig. 7e, f). However, $C_{12}$-$G^+$ did not fully suppress infection by *Z. tritici* or *M. oryzae*, and $C_{18}$-$NMe_3^+$ did not afford complete protection against rice blast disease (Fig. 7e, f). In contrast, $C_{18}$-$SMe_2^+$ vanquished symptom development at 75 and 100 $\mu$g ml$^{-1}$ in wheat and rice (Fig. 7e, f). A direct comparison of disease symptom formation at high concentrations revealed statistically significant improved protective effects of $C_{18}$-$SMe_2^+$ (Fig. 7g, h).

$C_{18}$-$SMe_2^+$ has a multiple MoA, which may underlie its plant protective activity. We also considered it possible that $C_{18}$-$SMe_2^+$ could trigger plant defense, so "alerting" the plant to potential pathogen attack. Initiating plant defense results in an early oxidative burst, including the production of hydrogen peroxide, which protects the plant from pathogen invasion[49]. We tested if MALCs induce this oxidative burst, by treating rice leaves with 100 $\mu$g ml$^{-1}$ $C_{12}$-$G^+$, $C_{18}$-$NMe_3^+$, and $C_{18}$-$SMe_2^+$, followed by staining with diaminobenzidine (DAB). This dye reacts with local $H_2O_2$, to give a brown precipitate indicative of a plant defense reaction[50]. Solvent-treated rice leaves showed no DAB precipitation (Fig. 7i, j; negative), whereas treatment with 15 mM salicylic acid resulted in brown colouration (Fig. 7i, j; positive). Both $C_{12}$-$G^+$ and $C_{18}$-$NMe_3^+$ treatment induced slight peroxide production after 6 h (Fig. 7i, j; both significantly higher than negative control), but a much stronger DAB reaction was found when leaves were sprayed with $C_{18}$-$SMe_2^+$ (Fig. 7i, j; Table 1). We tested if this activation of plant defense negatively impacts rice growth. Fourteen-day-old rice (Fig. 7k, pre-spray) was sprayed twice with solvent (Control) or 100 or 150 $\mu$g ml$^{-1}$ $C_{18}$-$SMe_2^+$, and grown for additional 14 days. Plants were harvested at

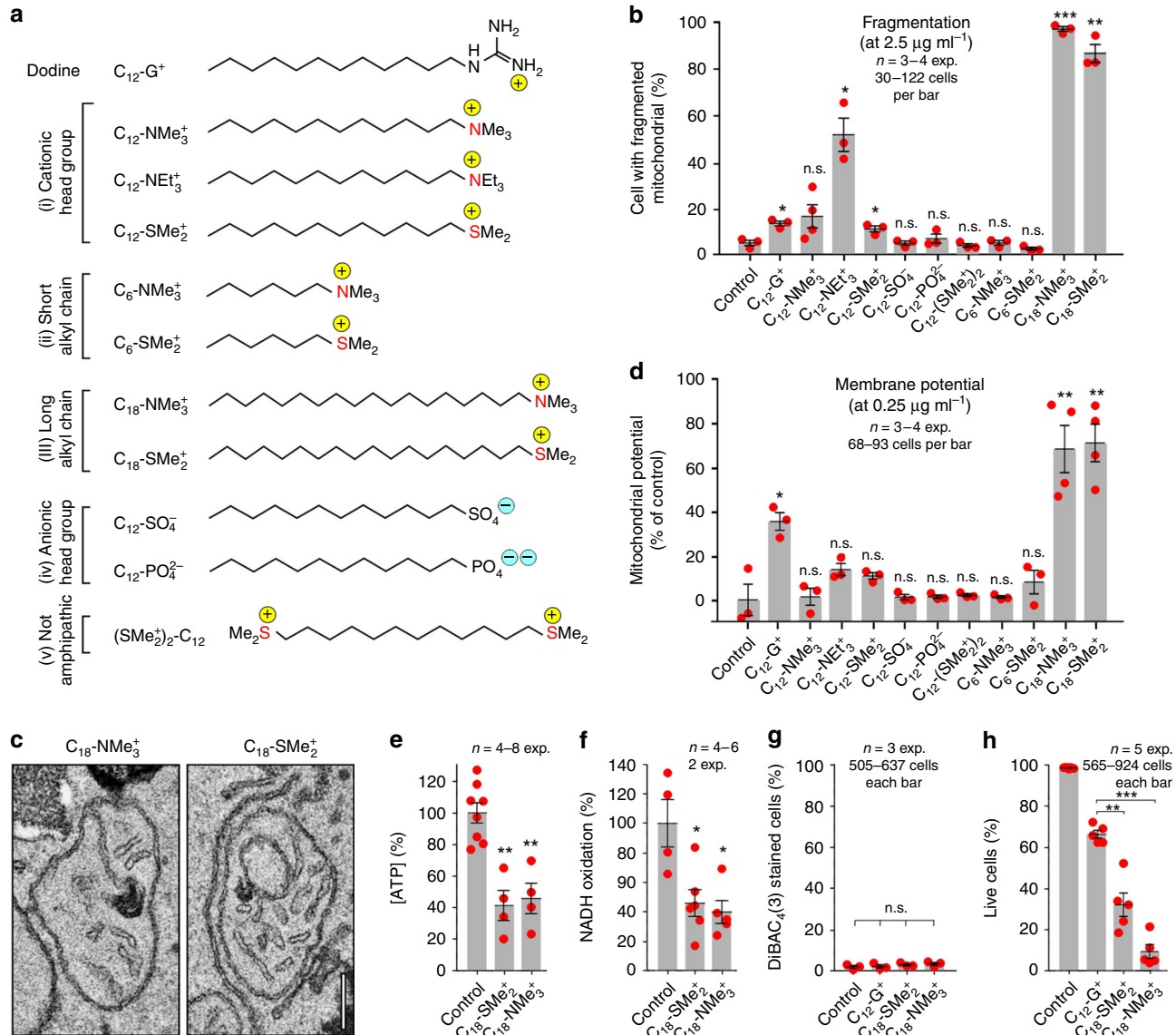

**Fig. 4 Effect of various MALCs and related compounds on mitochondria in *Z. tritici*. a** Structure of MALCs and related compounds. Ionic charges are indicated by "+" and "−". The counter ions for all lipophilic cations are: (1) acetate ($C_{12}$-$G^+$), (2) bromide ($C_{12}$-$NMe_3^+$, $C_{12}$-$NEt_2^+$, $C_6$-$NMe_3^+$, $C_{18}$-$NMe_3^+$), (3) iodide ($C_{12}$-$SMe_2^+$, $C_6$-$SMe_2^+$, $C_{18}$-$SMe_2^+$, $(SMe_2^+)_2$-$C_{12}$). **b** Effect of MALCs and related compounds on mitochondrial fragmentation in *Z. tritici* at 2.5 µg ml⁻¹. See also Supplementary Figs. 4a, 5. **c** Effect of various $C_{18}$-$NMe_3^+$ and $C_{18}$-$SMe_2^+$ on mitochondrial ultrastructure of mitochondria. Note swollen cristae. Scale bar = 0.2 µm. **d** Effect of various compounds on mitochondrial inner membrane potential, monitored using TMRM, in *Z. tritici* at a concentration of 0.25 µg ml⁻¹. **e** Bar chart showing the relative amount of ATP in cell extracts, pre-treated for 2 h with $C_{18}$-$SMe_2^+$ and $C_{18}$-$NMe_3^+$. The ATP concentration in cell extracts from solvent-only treated cells was set to 100%. ATP concentration was measured using a luciferase-based assay. **f** NADH oxidation measured in isolated mitochondria in the presence of $C_{18}$-$SMe_2^+$ and $C_{18}$-$NMe_3^+$. **g** Number of DiBAC₄(3)-positive cells (=depolarized cells), treated with solvent (Control), $C_{18}$-$SMe_2^+$, $C_{18}$-$NMe_3^+$ and $C_{12}$-$G^+$. Only cells that were negative for live/dead staining were included in the analysis. **h** Quantitative live/dead staining of cells after treatment with $C_{12}$-$G^+$, $C_{18}$-$SMe_2^+$, and $C_{18}$-$NMe_3^+$. Values (**b, d–h**) are shown as mean ± SEM. Red dots represent data points. Sample size *n* is indicated in each panel. Student's *t*-testing with Welch correction in (**b, d–h**) and one-way ANOVA (**g**); n.s. non-significant difference to control ($P > 0.05$; **b, d, g**); *$P = 0.0127$ (**b**, $C_{12}$-$G^+$), 0.0236 (**b**, $C_{12}$-$NEt_3^+$), 0.0391 (**b**, $C_{12}$-$SMe_2^+$), 0.0226 (**d**), 0.0428 (**f**, $C_{18}$-$SMe_2^+$), 0.0281 (**f**, $C_{18}$-$NMe_3^+$); **$P = 0.0024$ (**b**), 0.0061 (**d**, $C_{18}$-$NMe_3^+$), 0.0031 (**d**, $C_{18}$-$SMe_2^+$), 0.0037 (**e**, $C_{18}$-$SMe_2^+$), 0.0055 (**e**, $C_{18}$-$NMe_3^+$), and 0.0048 (**h**); ***$P < 0.0001$ (**b, h**). All *P*-values are two-tailed. See Supplementary Table 7 for experimental conditions. All source data are provided as a Source Data file.

42 days and fresh leaf/stem weight determined. This revealed no negative effect of $C_{18}$-$SMe_2^+$ on aerial biomass. In fact, $C_{18}$-$SMe_2^+$ slightly promoted growth (Fig. 7l).

Collectively, our results show that $C_{18}$-$SMe_2^+$ protects wheat and rice against fungal infection. This is most likely due to (i) induction of aggressive mROS in the pathogen, (ii) induction of fungal apoptosis (iii) alerting the plant defense system.

**$C_{18}$-$SMe_2^+$ is non-mutagenic and of low toxicity.** Low toxicity to humans and the environment is an important requirement for fungicides[3]. We set out to test the effect of $C_{18}$-$NMe_3^+$ and $C_{18}$-$SMe_2^+$ on human cells and water fleas, with the goal of comparing this data with $C_{12}$-$G^+$, currently used to control apple scab[27]. We tested the effect on mitochondria in human skin fibroblasts C109 and found that the $C_{18}$-alkyl chain cations

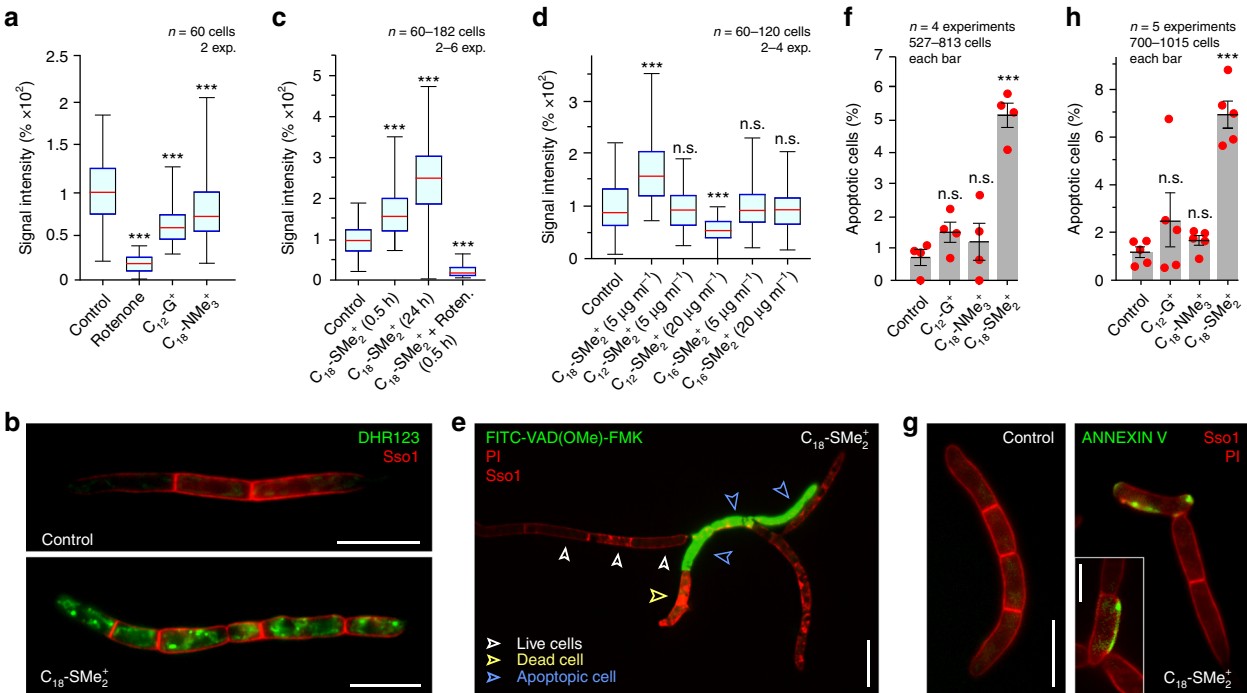

**Fig. 5 C$_{18}$-SMe$_2$$^+$-induced ROS production and programmed cell death in Z. tritici. a** Quantitative comparison of DHR-123 staining of mROS in *Z. tritici* cells, treated for 30 min with solvent (Control), the complex I inhibitor rotenone, or C$_{12}$-G$^+$ and C$_{18}$-NMe$_3$$^+$. **b** DHR-123 staining of mROS in *Z. tritici* cells, treated with solvent (Control) and C$_{18}$-SMe$_2$$^+$. The plasma membrane is labeled by mCherry-Sso1. Note that the red-fluorescence images were scaled identically. Scale bars = 10 μm. **c** mROS production in *Z. tritici* in presence of solvent control (Control), 30 min treatment with C$_{18}$-SMe$_2$$^+$ (C$_{18}$-SMe$_2$$^+$, 0.5 h), and 24 h treatment with C$_{18}$-SMe$_2$$^+$ (C$_{18}$-SMe$_2$$^+$, 24 h), and co-treatment with rotenone and C$_{18}$-SMe$_2$$^+$ (C$_{18}$-SMe$_2$$^+$+Roten., 0.5 h). **d** mROS production in *Z. tritici* after treatment with solvent (Control), C$_{12}$-SMe$_2$$^+$, C$_{16}$-SMe$_2$$^+$, and C$_{18}$-SMe$_2$$^+$, at 5 μg ml$^{-1}$ or 20 μg ml$^{-1}$. See also Supplementary Fig. 6. **e** Apoptotic *Z. tritici* cells, detected by CaspACE FITC-VAD-fmk. Healthy cells exclude the dye (open white arrowheads), whereas apoptotic cells are filled with green-fluorescence (open blue arrowheads). Post-apoptotic cells that contain propidium iodide were not included in the analysis (yellow open arrowhead). The plasma membrane is visualized by the marker mCherry-Sso1 (red). Scale bar = 10 μm. **f** Number of early apoptotic *Z. tritici* cells (CaspACE FITC-VAD-fmk-positive, but propidium iodide-negative), treated with the solvent, C$_{12}$-G$^+$, C$_{18}$-NMe$_3$$^+$, and C$_{18}$-SMe$_2$$^+$. **g** Apoptotic *Z. tritici* cells, detected by Annexin-V-fluorescein staining. Healthy cells are unstained (left panel), whereas apoptotic cells expose phosphatidylserine and show green-fluorescence at their plasma membrane (right panel; plasma membrane in red, labeled with mCherry-Sso1). Post-apoptotic cells were identified by double staining with propidium iodide and excluded from the analysis. Scale bars = 10 μm (left panel), 5 μm (insert, right panel). **h** Number of early apoptotic *Z. tritici* cells (Annexin-V-fluorescein-positive, but propidium iodide-negative), treated with the solvent control, C$_{12}$-G$^+$, C$_{18}$-NMe$_3$$^+$ and C$_{18}$-SMe$_2$$^+$. **a, c, d** Whiskers' plots, with 25th/75th percentiles (blue lines), medians (red line), and minimum and maximum (whiskers ends). Values **f, h** are shown as mean ± SEM. Red dots represent data points. Sample size *n* is indicated in each panel. Non-parametric Mann–Whitney testing (**a, c, d**); n.s. non-significant difference to control (*P* > 0.05) (**d**), ***P < 0.0001 (**a, c, d**). Student's *t*-testing with Welch correction (**f, h**); n.s. non-significance (**f, h**) and ***P = 0.0002 (**f, h**). All *P*-values are two-tailed. See Supplementary Table 7 for experimental conditions. All source data are provided as a Source Data file.

C$_{18}$-NMe$_3$$^+$ and C$_{18}$-SMe$_2$$^+$ induced changes in mitochondrial morphology in fungi at 1.12–1.25 μg ml$^{-1}$, whereas human mitochondrial morphology was ~28–32-times less sensitive to both MALCs (Fig. 8a; Table 1). Moreover, the human cell mitochondrial potential was much less sensitive to MALC treatment (Fig. 8b), with C$_{18}$-NMe$_3$$^+$ being ~90-times and C$_{18}$-SMe$_2$$^+$ ~50-times more specific for fungi than for humans (Table 1). Both MALCs were ~1.4-times less toxic to mitochondrial morphology than C$_{12}$-G$^+$ (Table 1), but inhibited the mitochondrial potential in C109 cells at slightly lower doses than C$_{12}$-G$^+$ (Table 1). We next tested the cytotoxicity of C$_{18}$-NMe$_3$$^+$, C$_{18}$-SMe$_2$$^+$, and C$_{12}$-G$^+$ using a tetrazolium salt assay (MTT-assay), used to assess cell viability and proliferation in cultured cells[51] on human C109 skin fibroblasts and HepG2 hepatoblastoma cells at various MALC concentrations. Both C$_{18}$-NMe$_3$$^+$ and C$_{18}$-SMe$_2$$^+$ were clearly less cytotoxic than C$_{12}$-G$^+$ (Fig. 8c; Table 1 and Supplementary Fig. 9).

Next, we tested the effect of C$_{12}$-G$^+$, C$_{18}$-NMe$_3$$^+$, and C$_{18}$-SMe$_2$$^+$ on the water flea *Daphnia magna*. This freshwater crustacean is well-established as a reporter organism in toxicity tests[52]. Firstly, we treated *Daphnia magna* with 1 μg ml$^{-1}$ of C$_{12}$-G$^+$, C$_{18}$-NMe$_3$$^+$, and

C$_{18}$-SMe$_2$$^+$ for 30 min, followed by monitoring mitochondrial potential, with TMRM. This treatment did not affect the crustacean motility. Water fleas incubated with solvent control showed red-fluorescence, suggesting healthy mitochondria (Fig. 8d, Control). In contrast, C$_{12}$-G$^+$-treated organisms lost signal (Fig. 8d, C$_{12}$-G$^+$), indicating that even low amounts of C$_{12}$-G$^+$ rapidly affected mitochondrial respiration. Under these conditions, C$_{18}$-NMe$_3$$^+$ also slightly affected mitochondrial respiration, whereas C$_{18}$-SMe$_2$$^+$ was without any obvious effect (Fig. 8d). Next, we treated water fleas for 24 h with various concentrations of the three MALCs and monitored motility behavior and "escape response", as an indicator of mortality[53]. Here, C$_{12}$-G$^+$ immobilized all *Daphnia magna* at ~1 μg ml$^{-1}$, while C$_{18}$-NMe$_3$$^+$ and C$_{18}$-SMe$_2$$^+$ had little effect (Fig. 8e, Supplementary Movie 1). 50% *Daphnia magna* immobilization was detected at ~2 μg ml$^{-1}$ (C$_{18}$-NMe$_3$$^+$) and ~3.6 μg ml$^{-1}$ (C$_{18}$-SMe$_2$$^+$; Table 1; Fig. 8e). Thus, C$_{18}$-SMe$_2$$^+$ showed 1.5-times lower toxicity in water fleas than C$_{18}$-NMe$_3$$^+$, and is ~7.8-times less toxic than C$_{12}$-G$^+$.

C$_{12}$-G$^+$ was shown to be non-mutagenic in AMES tests[54], a widely-used genotoxicity assay that monitors DNA mutations that revert auxotrophic bacterial reporter strains to prototrophy[55,56].

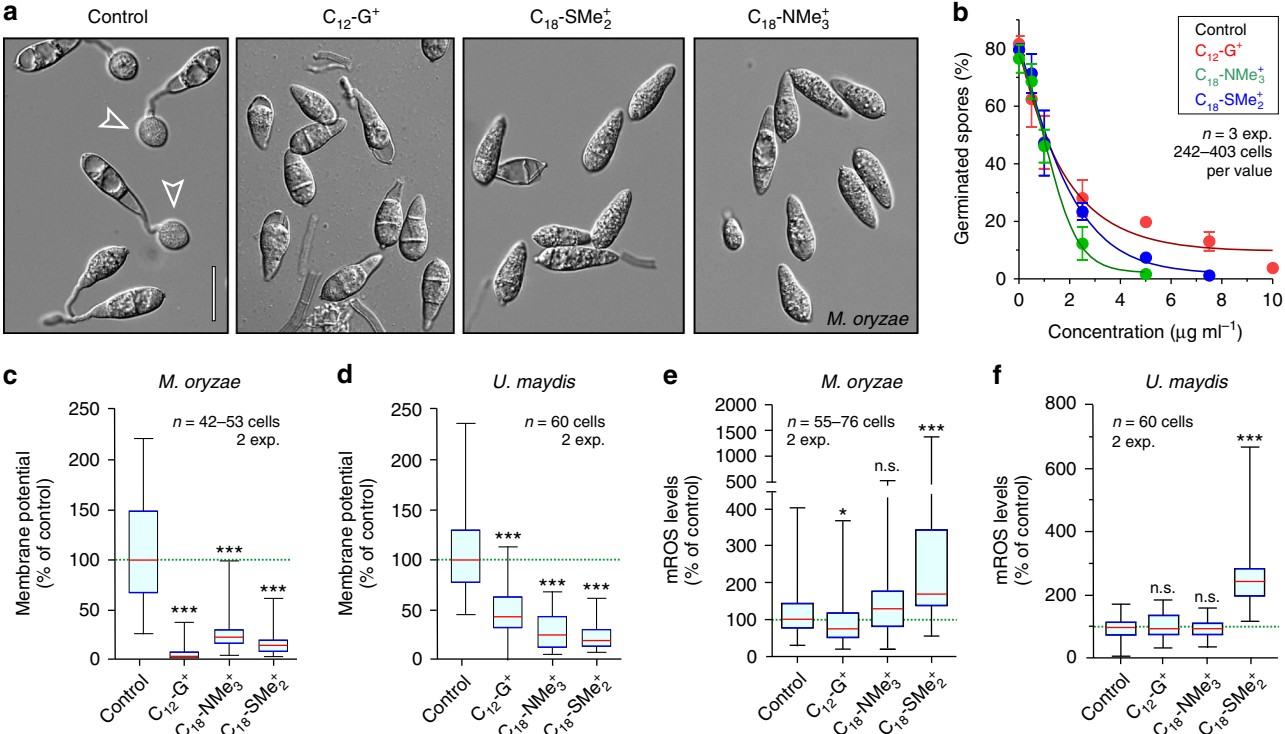

**Fig. 6 Effect of $C_{12}$-$G^+$, $C_{18}$-$NMe_3^+$, and $C_{18}$-$SMe_2^+$ on the rice blast fungus *M. oryzae* and the corn smut fungus *U. maydis*. a** Effect of treatment of *M. oryzae* conidia with the solvent (control), $C_{12}$-$G^+$, $C_{18}$-$SMe_2^+$, and $C_{18}$-$NMe_3^+$. In control experiments, appressoria are formed (Control, open arrowheads), whereas all MALCs inhibit germination of *M. oryzae* conidia at high concentrations. Scale bar = 20 μm. **b** Effect of $C_{12}$-$G^+$, $C_{18}$-$NMe_3^+$, and $C_{18}$-$SMe_2^+$ on germination of *M. oryzae* conidia. See Table 1 for $EC_{50}$ values. **c** Effect of $C_{12}$-$G^+$, $C_{18}$-$SMe_2^+$, and $C_{18}$-$NMe_3^+$ on mitochondrial potential in *M. oryzae*. Values are background-corrected TMRM fluorescence intensity values. See also Supplementary Fig. 7b. **d** Effect of $C_{12}$-$G^+$, $C_{18}$-$SMe_2^+$, and $C_{18}$-$NMe_3^+$ on mitochondrial potential in *U. maydis*. Values are background-corrected TMRM fluorescence intensity values. See also Supplementary Fig. 7d. **e** Effect of $C_{12}$-$G^+$, $C_{18}$-$SMe_2^+$, and $C_{18}$-$NMe_3^+$ on the mROS levels in *M. oryzae*. Only $C_{18}$-$SMe_2^+$ induces significantly mROS levels. **f** Effect of $C_{12}$-$G^+$, $C_{18}$-$SMe_2^+$, and $C_{18}$-$NMe_3^+$ on the mROS levels in *U. maydis*. Only $C_{18}$-$SMe_2^+$ induces significantly mROS levels. See also Supplementary Fig. 7e. Values (**b**) are shown as mean ± SEM, with sample size $n = 3$. Non-linear regression curves were calculated as dose-response inhibition (four parameters) in Prism5. **c**–**f** Whiskers' plots, with 25th/75th percentiles (blue lines), medians (red line), and minimum and maximum (whiskers ends). Green lines indicate level of control median. Sample size $n$ is indicated in each panel. Non-parametric Mann–Whitney testing (**a**, **c**, **d**); n.s. non-significant difference to control ($P > 0.05$) (**d**), ***$P < 0.0001$ (**a**, **c**, **d**). Student's *t*-testing with Welch correction (**f**, **h**); n.s. non-significance (**f**, **h**) and ***$P = 0.0002$ (**f**, **h**). All *P*-values are two-tailed. See Supplementary Table 7 for experimental conditions. All source data are provided as a Source Data file.

While our results indicate that $C_{18}$-$SMe_2^+$ has lower toxicity than $C_{12}$-$G^+$ in human cells and water fleas, we asked if this MALC has mutagenic activity. Gentronix Ltd. (Cheshire, UK) assessed the mutagenic potential of $C_{18}$-$SMe_2^+$, using AMES testing according to OECD (Organization for Economic Co-operation and Development) guideline 471. This approach revealed no mutagenic potential of $C_{18}$-$SMe_2^+$ in five tested reporter strains, both with or without metabolic activation by a liver S9 fraction (Fig. 8f, g and Supplementary Tables 4, 5). However, $C_{18}$-$SMe_2^+$ displayed bactericidal activity at doses over 35 μg ml$^{-1}$.

## Discussion

We need new fungicides to protect our calorie crops from fungal disease and thus ensure food security. The challenge is to discover antifungals with low environmental impact. Moreover, such chemistries must provide sustained control over the seasons, ideally having multi-site modes of action that cannot be easily overcome by the emergence of resistance[3]. Mitochondria are valuable targets for fungicide development, as these organelles provide cellular ATP, but also control lipid homeostasis and programmed fungal cell death[14,18]. In this study, we synthesized and investigated a group of MALCs, including variants with a dimethylsulfonium head group. To our knowledge, such compounds have not been tested against fungi. We report that a long

chain dimethylsulfonium cation, $C_{18}$-$SMe_2^+$ is specific for fungi, shows no genotoxicity, is less toxic to human fibroblasts and *Daphnia magna* than the existing fungicide dodine and shows no detectable phytotoxicity in crop plants. We show that $C_{18}$-$SMe_2^+$ combats fungal pathogens by (i) inhibiting oxidative phosphorylation, (ii) inducing damaging mROS, (iii) triggering fungal apoptosis, (iv) initiating plant defense. This multi-site MoA makes it unlikely that pathogens will readily develop resistance to $C_{18}$-$SMe_2^+$.

Lipophilic cations are known for their bacterial toxicity[25] and, more recently, gained recognition as anti-fungal compounds in medicine[26]. Amongst the biocidal lipophilic cations are the MALCs, which combine a cationic head group with a long *n*-alky chain. This simple amphipathic organization suggests that MALCs insert into membranes[25] and several studies report a primary activity at the plasma membrane or on the cell wall[26,29]. However, early reports suggested that $C_{12}$-$G^+$ inhibits vital metabolic enzymes, thus exerting its primary toxic effect inside the fungal cell[30,31]. Such activity inside the cell requires passive diffusion through the plasma membrane into the cell. Membrane permeability depends on the lipophilic nature of a molecule, with positive Log*P* values indicating easier passage through bio-membranes[21]. The estimated Log*P* value for $C_{12}$-$G^+$ is 2.26, which is similar to that of the mitochondrial dye R123

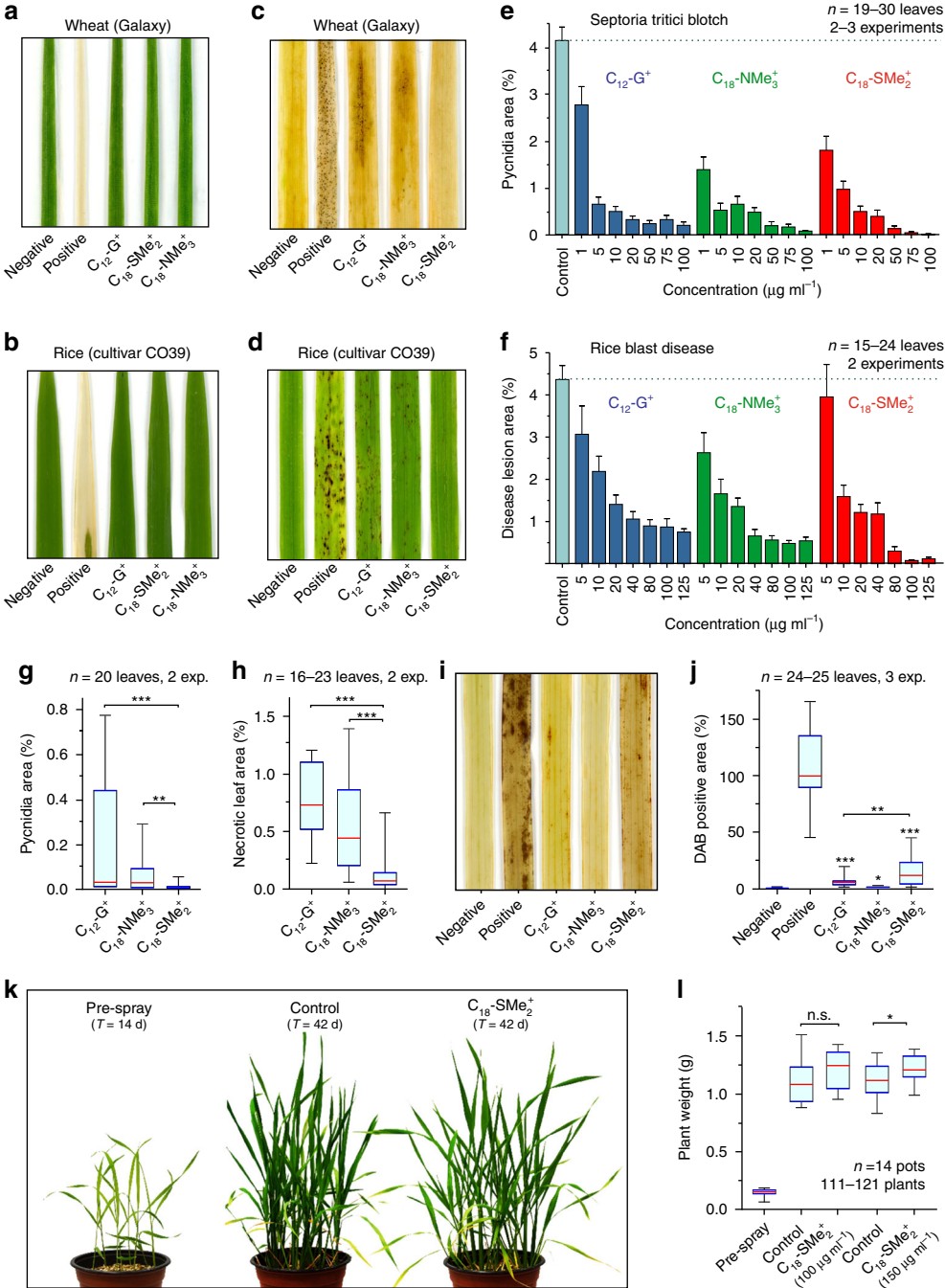

**Fig. 7 Phytotoxicity and plant protection by $C_{12}$-$G^+$, $C_{18}$-$NMe_3^+$, and $C_{18}$-$SMe_2^+$ in wheat and rice. a**, **b** Phytotoxicity of $C_{12}$-$G^+$, $C_{18}$-$NMe_3^+$, and $C_{18}$-$SMe_2^+$ in wheat (**a**) and rice (**b**). See also Supplementary Fig. 8a. **c** Septoria wheat blotch disease symptoms on wheat leaves, pre-treated with $C_{12}$-$G^+$, $C_{18}$-$NMe_3^+$, and $C_{18}$-$SMe_2^+$. Dark spots represent melanised pycnidia. See also Supplementary Fig. 8b. **d** Rice blast disease symptoms on rice leaves. Dark areas represent necrotic lesions. See also Supplementary Fig. 8b. **e** Disease symptoms in wheat after treatment with various concentrations of $C_{12}$-$G^+$, $C_{18}$-$SMe_2^+$, and $C_{18}$-$NMe_3^+$. Note that $C_{18}$-$SMe_2^+$ provides greatest protection against fungal infection. **f** Disease symptoms in rice after treatment with various concentrations of $C_{12}$-$G^+$, $C_{18}$-$SMe_2^+$, and $C_{18}$-$NMe_3^+$. Note that $C_{18}$-$SMe_2^+$ provides greatest protection against fungal infection. **g** Comparison of disease symptoms in wheat, pre-treated with $C_{12}$-$G^+$, $C_{18}$-$NMe_3^+$, and $C_{18}$-$SMe_2$. **h** Comparison of disease symptoms in rice, pre-treated with $C_{12}$-$G^+$, $C_{18}$-$NMe_3^+$, and $C_{18}$-$SMe_2^+$. **i** DAB staining of rice leaves at 6 h after treatment with $C_{12}$-$G^+$, $C_{18}$-$NMe_3^+$, and $C_{18}$-$SMe_2^+$. **j** Quantitative comparison of DAB precipitation in rice leaves at 6 h after spraying with $C_{12}$-$G^+$, $C_{18}$-$NMe_3^+$, and $C_{18}$-$SMe_2^+$. Note that no DAB reaction was seen at 2 h after treatment. **k** Effect of $C_{18}$-$SMe_2^+$ on rice growth (Pre-spray: 14 days old; left pot), control and $C_{18}$-$SMe_2^+$-treated plants: 42 days old (middle and right pot). **l** Fresh weight of rice plants, grown for 14 days (Pre-spray) and 42 days (Control and $C_{18}$-$SMe_2^+$). Values (**e**, **f**) are shown as mean ± SEM; sample sizes are indicated in graphs. **g**, **h**, **j**, **l** Whiskers' plots, with 25th/75th percentiles (blue lines), medians (red line), and minimum and maximum (whiskers ends). Sample size $n$ is indicated in each panel. Non-parametric Mann–Whitney testing in (**g**, **h**, **j**, **l**); statistical comparisons are indicated by brackets; in absence of brackets comparison with negative control is shown; n.s. non-significant difference ($P = 0.1029$) (**l**); *$P = 0.0102$ (**j**) and $P = 0.0409$ (**l**); **$P = 0.0017$ (**g**) and $P = 0.0052$ (**j**); ***$P = 0.001$ (**g**) and $P < 0.0001$ (**h**, **j**). All $P$-values are two-tailed. See Supplementary Table 7 for experimental conditions. All source data are provided as a Source Data file.

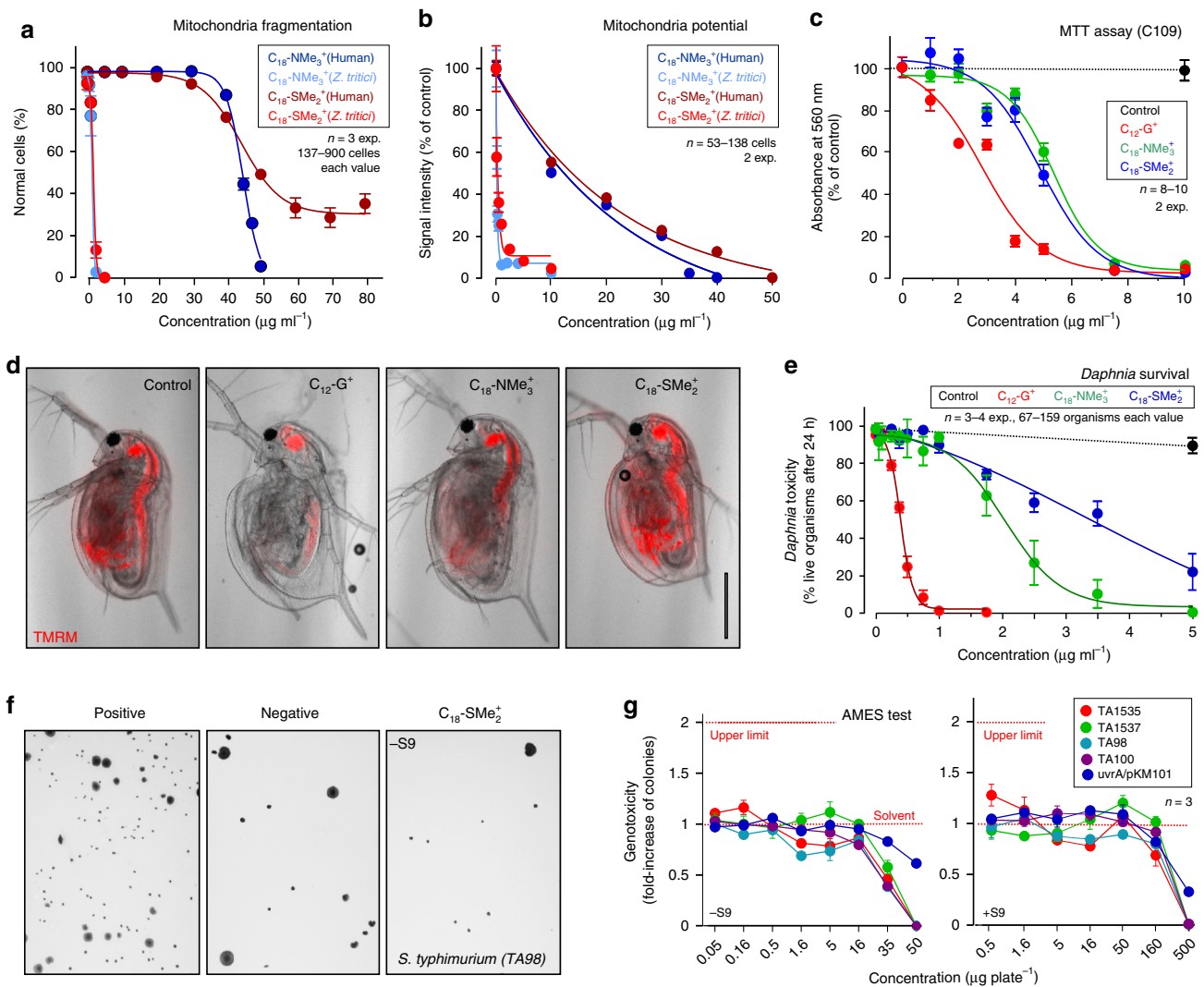

**Fig. 8 Toxicity of MALCs in human C109 cells and *Daphnia magna* and genotoxicity of $C_{18}$-$SMe_2^+$. a** Concentration-dependent fragmentation of mitochondria in human C109 fibroblasts (Human) and *Z. tritici* cells, treated with $C_{18}$-$SMe_2^+$ and $C_{18}$-$NMe_3$. See Table 1 for $EC_{50}$ values. **b** Concentration-dependent depolarization of mitochondria in human C109 fibroblasts (Human) and *Z. tritici* cells, treated with $C_{18}$-$SMe_2^+$ and $C_{18}$-$NMe_3^+$. See Table 1 for $EC_{50}$ values. **c** MTT test for cell viability and metabolic activity in human C109 fibroblasts, treated for 24 h with various concentrations of $C_{12}$-$G^+$, $C_{18}$-$SMe_2^+$ and $C_{18}$-$NMe_3^+$. See Table 1 for $EC_{50}$ values and Supplementary Fig. 9 for MTT assay with HepG2 cells. **d** Mitochondrial potential, visualized by TMRM, in living *Daphnia magna* after incubation with solvent, $C_{12}$-$G^+$, $C_{18}$-$SMe_2^+$, and $C_{18}$-$NMe_3^+$. Note that water fleas are still alive but immobilized for microscopy. Scale bar = 500 μm. **e** Mortality in *Daphnia magna* after 24 h treatment with solvent (Control) and $C_{12}$-$G^+$, $C_{18}$-$SMe_2^+$, and $C_{18}$-$NMe_3^+$ at various concentrations. Live water fleas showed internal movement of gills, or heart-beat or swimming behavior. See Table 1 for $EC_{50}$ values. **f** Growth of mutants of the *S. typhimurium* genotoxicity reporter strain TA98 on agar plates supplemented with 1 μg ml$^{-1}$ of 2-nitrofluorene (Positive), 100 μl of solvent dimethyl sulfoxide (Negative) and 0.1 μg ml$^{-1}$ $C_{18}$-$SMe_2^+$. Note that growth indicates a frame shift mutation that reverses the strain from auxotrophy to prototrophy. **g** Test for genotoxicity of $C_{18}$-$SMe_2^+$ at various concentrations. –S9: no metabolic activation; +S9: metabolic activation by rat liver membranes. Note that the Ames test result is considered "negative" (=no mutagenic potential), if the number of prototrophic mutants stays is less than 2-fold increased (indicated by upper red dotted line). See also Supplementary Tables 4, 5. Values (**a**–**c**, **e**, **g**) are shown as mean ± SEM; sample sizes are indicated in graphs. Non-linear regression curves in (**a**–**c**, **e**) were calculated as dose-response inhibition (four parameters) in Prism5. See Supplementary Table 7 for experimental conditions. All source data are provided as a Source Data file.

($LogP_{R123} = 2.15$) or the mitochondrial potential indicator TMRM ($LogP_{TMRM} = 2.74$), and comparable to the strobilurin fungicide azoxystrobin ($LogP_{azoxystrobin} = 2.97$; Supplementary Table 1). Thus, it is very likely that $C_{12}$-$G^+$ diffuses passively through the plasma membrane. Once the positively-charged MALC reaches the negatively-charged mitochondrial matrix, they, as other lipophilic cations[22], likely concentrate up to 100–1000-times and insert into the IMM. Our results support this concept. While high concentrations of $C_{12}$-$G^+$ affect the organization of the plasma

membrane ($EC_{50} \sim 50$ μg ml$^{-1}$ in *Z. tritici*), much lower concentrations induce mitochondrial fragmentation, alter cristae organization and inhibit oxidative phosphorylation. We confirm these findings for $C_{12}$-$G^+$ and two MALCs ($C_{18}$-$NMe_3^+$ and $C_{18}$-$SMe_2^+$) in *Z. tritici*, *M. oryzae*, and *U. maydis*. Thus, we conclude that the primary target of MALCs in fungi are mitochondria, where they interfere with synthesis of ATP.

Several lines of evidence suggest that MALCs affect mitochondrial respiration and oxidative phosphorylation. Firstly, we

show that MALC treatment reduces fluorescence of the membrane potential indicator TMRM. This suggests that mitochondria are depolarized, which is expected to reduce the proton-motive force, required to synthesize ATP[10]. Secondly, we show that MALC treatment indeed inhibits ATP production and oxygen consumption. As oxygen receives the electrons at the end of the respiration chain, MALCs appear to affect electron transfer through the respiration chain. Thirdly, we provide evidence that MALCs inhibit NADH oxidation, which generates the electrons for respiration. In filamentous fungi, such NADH oxidation occurs at complex I, which contains fungal-specific subunits[13,14], and at alternative fungal-specific NADH dehydrogenases[11,12]. Thus, the specificity of MALCs on fungi may relate to the inhibition of these unique respiratory enzymes. Alternatively, MALCs could have additional effects in mitochondria, such as a protonophoric activity, reported for the lipophilic cation $C_{12}TPP^+$ in human cells[23], or impact on membrane fluidity, shown to affect respiration complex assembly and electron transfer[13,57,58] (Fig. 9). More physiological experiments are needed to elucidate the detailed mechanism of action of MALCs.

The most effective antifungal MALC against pathogen infection is $C_{18}-SMe_2^+$. We consider it likely that this improved protective performance is due to the multiple ways in which $C_{18}-SMe_2^+$ "attacks" the fungal pathogen: (i) like other MALCs, it inhibits oxidative phosphorylation, so depriving the fungal cell of ATP; (ii) it induces mROS production, which is expected to cause oxidative damage to mitochondrial lipids and proteins[59]; (iii) $C_{18}-SMe_2^+$-induced mROS propels the pathogen down an irreversible apoptotic pathway[16,59], which is indicated by the induction of metacaspase activity and the exposure of phosphatidylserines at the plasma membrane, both markers of programmed cell death[44,47]; (iv) $C_{18}-SMe_2^+$ triggers the formation of hydrogen peroxide in rice plants, being indicative of the oxidative burst which triggers early plant defense responses[49]. Thus $C_{18}-SMe_2^+$ likely alerts the plant against pathogen attack, which decreases the chance of fungal infection.

Most of the improved anti-fungal performance of $C_{18}-SMe_2^+$ is likely due to its mROS-inducing activity (Fig. 9). In the presence of rotenone, $C_{18}-SMe_2^+$ no longer causes mROS production, suggesting that $C_{18}-SMe_2^+$ acts on respiratory chain complex I. Here, the molecule may associate with the hydrophilic core of the enzyme complex, where the electrophilic dimethylsulfonium moiety could redirect electrons from the respiration chain[60]. Indeed, direct binding of lipophilic cations to the proteins of respiration complexes was shown previously[61]. Such direct interaction may require a long alkyl chain, which could explain why the slightly shorter $C_{16}-SMe_2^+$ compound does not induce mROS in Z. tritici. However, other explanations are possible, and we do not exclude a more unspecific effect of $C_{18}-SMe_2^+$ on the organization of the respiration chain, which affects complex I most, thus inducing mROS. We also speculate that mROS induction by $C_{18}-SMe_2^+$ induces an oxidative burst in rice plants. However, further studies are needed to elucidate the role of $C_{18}-SMe_2^+$ in triggering plant defense. Importantly, we report here that the $C_{18}-SMe_2^+$ also induces mROS in M. oryzae and U. maydis. Thus, this important feature of $C_{18}-SMe_2^+$ is of general relevance in fighting fungal pathogens.

We are in an arms-race against crop pathogens to secure our future food security. The rapid development of resistance in market leader chemistries makes the identification of new fungicides a priority[4]. More sustainable fungicides, combining lower risk of resistance development and a safe toxicity profile to humans and the environment are the current drivers in crop protection research. A good example of such a recent antifungal is amphiphilic kanamycin K20, which combines low toxicity with broad range anti-fungal activity, both in human and plant

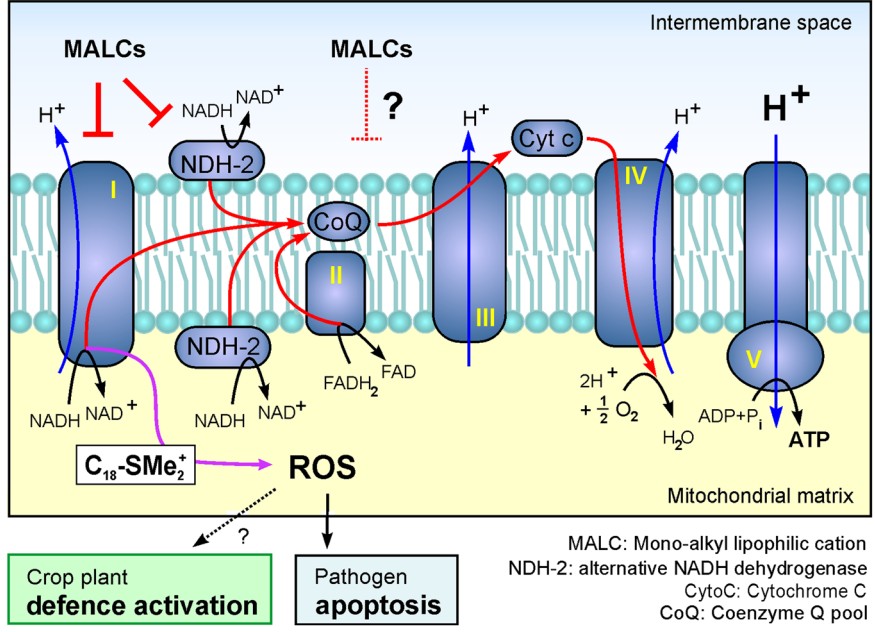

**Fig. 9 Model of the effect of MALCs on fungal oxidative phosphorylation.** Our results from several assays indicate that all MALCs inhibit oxidative phosphorylation. This could be a consequence of MALC insertion into the inner membrane, followed by inhibition of NADH oxidation and electron transfer through the electron transport chain. NADH oxidation in fungi involves unique enzymes (alternative NADH dehydrogenases, indicated by NDH-2) and respiratory complex I, which contains fungal-specific subunits. Thus, the higher toxicity of MALCs to fungi may be linked to an activity on these fungal-specific compounds. However, additional effects on IMM fluidity and organization of the respiration complexes cannot be excluded (indicated by "?"). In addition, $C_{18}-SMe_2^+$ induces mROS production at complex I, suggesting that it may bind to or interact with this enzyme complex. Aggressive mROS damages lipids and proteins in the inner membrane, but also activates apoptosis in the pathogen cell. Moreover, $C_{18}-SMe_2^+$-induced mROS production may participate in a mild activation of the plant defense system, evidenced by a slight oxidative burst 6 h post-treatment in rice leaves.

pathogens[62]. Like MALCs, this molecule is cationic and carries a lipophilic moiety. However, its overall lipophilicity is very low ($LogP = -3.06$), and similar to ATP ($LogP = -3.39$) or glucose ($LogP = 2.26$; Supplementary Table 1). Thus, K20 is most likely not passing passively through bio-membranes. Instead, it appears to exert its cytotoxic activity at the fungal plasma membrane (overview in the ref. [63]). New fungicides should (i) be active against crop-destroying pathogens (ii) should target a fundamental process at multiple sites to reduce resistance development, (iii) be of low toxicity to humans and the environment[3,8]. We show here that MALCs meet these criteria. Firstly, MALCs are effective against Septoria tritici leaf blotch pathogen Z. tritici, rice blast fungus M. oryzae and corn smut fungus U. maydis. Collectively, these pathogens challenge crops that provide two thirds of calories in human diet[64], with Septoria tritici leaf blotch alone causing ~1.4 billion euros wheat loss in central Europe per year[32]. Secondly, the MALCs target the fungal respiration chain, which is an ideal target for fungicides[8,14]. MALCs do this in multiple ways, which is expected to reduce the risk of resistance development. Furthermore, we demonstrate here that the sulfonium head group in $C_{18}$-$SMe_2^+$ confers multiple fungicidal activities (mROS, apoptosis, and plant defense induction), which, in combination, represent multiple MoAs. The concentration by which $C_{18}$-$SMe_2^+$ exert their activity varies between our experiment set-ups. This is expected, as pathogen cells in liquid culture, or even naked human culture cells, are more sensitive to a given concentration of MALCs than fungi on the surface of plants, where the waxy cuticle likely absorbs lipophilic fungicides[65].

However, in almost all aspects tested, $C_{18}$-$SMe_2^+$ is superior in performance and shows less toxicity than $C_{12}$-$G^+$, currently used as the fungicide Syllit (dodine; Supplementary Table 6). More characterization of $C_{18}$-$SMe_2^+$ is needed, which should include more extended toxicity tests, but also the stability and bio-degradability of $C_{18}$-$SMe_2^+$. However, we propose that MALCs collectively should be considered in pathogen control strategies. In particular $C_{18}$-$SMe_2^+$ holds significant potential as a prospective fungicide.

## Methods

**Biological material**. Fungi: Z. tritici wild type strain IPO323 (CBS 115943) was obtained from the Fungal Biodiversity Center, Utrecht, Netherlands (http://www.cbs.knaw.nl). Z. tritici strains IPO323_eGFP-Sso1 and IP0323_Acd1-ZtGFP were described previously (for references see Supplementary Table 2) and can be obtained from the laboratory of the first author. The U. maydis wild type strain FB1 (CBS 132774) was provided by R. Kahmann, MPI Marburg, Marburg, Germany, and can also be obtained from the Fungal Biodiversity Center, Utrecht, Netherlands (http://www.cbs.knaw.nl). The M. oryzae wild type strain Guy11 (FGSC 9462) was provided by N. Talbot, Sainsbury Laboratory, Norwich, UK; it can also be requested from The Fungal Genetics Stock Center, Manhattan, KS, USA (http://www.fgsc.net/). Strain IPO323_mCh-ZtSsoI was generated by transforming plasmid pHmCherrySso1[33] into WT IPO323 using A. tumefaciens-mediated transformation[66]. Ustilago maydis strain FB1GSso1 was generated by ectopic integration of plasmid poGSso1[67] into strain FB1. Both strains can be obtained from the laboratory of the first author. Genotypes of all strains and plasmids are in Supplementary Table 2; experimental strain usage is in Supplementary Table 3.

Zooplankton: The water flea D. magna was obtained from the Northampton Reptile Center, Northampton, UK (https://www.reptilecentre.com/).

Bacteria: Salmonella typhimurium LT2 strains TA1535, TA1537, TA98, TA100, and Escherichia coli WP2 strain uvrA/pKM101 were provided by Gentronix Ltd., Cheshire, UK.

Mammalian cell culture cells: Human skin fibroblasts (C109) were provided by H. Waterham, University of Amsterdam, NL, and human hepatoblastoma cells (HepG2, HB-8065) were obtained from ATCC, Virginia, USA (http://www.atcc.org).

**Fungal growth conditions**. Solid media cultures Z. tritici were grown at 18 °C on YPD plates (yeast extract, 10 g l⁻¹; peptone, 20 g l⁻¹; glucose, 20 g l⁻¹; agar, 20 g l⁻¹). Liquid cultures were grown in YG medium (yeast extract, 10 g l⁻¹; glucose, 30 g l⁻¹) at 18 °C for 48 h, 200 rpm. U. maydis strains were grown on complete medium (CM) agar plates (1% (w v⁻¹) glucose) at 28 °C for 24–48 h. Liquid cultures were grown in

CM 1% (w v⁻¹) glucose, at 28 °C for 12 h, 200 rpm. M. oryzae strain Guy 11 conidia were plated onto CM agar plates (see above) at 25 °C for 10–15 days, spores by washing with sterile water, followed by filtration through two layers of sterile Miracloth (Calbiochem, Nottingham, UK) and centrifugation at 6000 rpm for 5 min. The spore pellet was re-suspended in sterile water and, for spore germination assays, aliquots of cell suspension were placed onto hydrophobic glass coverslips (Menzel-Glässer, Thermo Fisher Scientific, Loughborough, UK) and incubated in a wet chamber for 30 min at 25 °C.

**Fluorescence microscopy**. Fluorescence microscopy was performed as described[33]. Cells were observed using a motorized inverted microscope (IX81/IX83; Olympus, Hamburg, Germany), equipped with a PlanApo 100×/1.45 Oil TIRF, UPlanSApo 60×/1.35 Oil or a 4× UPlanFLN 0.13 objective (Olympus). The fluorescently-labeled proteins or fluorescent dyes were excited using a VS-LMS4 Laser Merge System with solid-state lasers (488 nm 75 mW and 561 nm 75 mW; Visitron Systems, Puchheim, Germany). For FRAP experiments a 405 nm/60 mW diode laser, coupled to the light path by an OSI-IX 71 adapter (Visitron), controlled by an UGA-40 controller (Rapp OptoElectronic GmbH, Hamburg, Germany) and VisiFRAP 2D FRAP control software for Meta Series 7.8.× (Visitron) was used. Images were captured using a CoolSNAP HQ2 camera (Photometrics/Roper Scientific, Tucson, USA). All parts were controlled by the software package Meta-Morph (Molecular Devices, Wokingham, UK). For live cell imaging of human skin fibroblasts a controlled-temperature chamber and objective was used at 37 °C.

**Electron microscopy**. Transmission electron microscopy on fungal cells was as described[68], using a JEOL JEM 1400 transmission electron microscope, operated at 120 kV. Images were taken with a digital camera (ES 1000 W, Gatan, Abingdon, UK). Mitochondria in human skin fibroblasts were visualized after fixation in 0.5% glutaraldehyde/0.2 M PIPES buffer, pH 7.2, for 15 min at room temperature (RT). Samples were centrifuged at $17,000 \times g$ for 10 min, washed with buffer and post-fixed for 1 h, in 1% osmium tetroxide in 0.1 M sodium cacodylate buffer, pH 7.2, then dehydrated and embedded in Durcupan resin (Sigma-Aldrich, Poole UK). Sixty nanometer ultrathin sections were collected on pioloform-coated 100-mesh copper EM grids (Agar Scientific), contrasted with lead citrate and analysed as described above.

**Statistical analysis and data presentation**. All measurements used 14-bit image files, analysed by the software MetaMorph. Data calculations were performed in Excel (Microsoft, Redmond, USA) or Prism5 (GraphPad Software, San Diego, USA). All statistical testing and non-linear and linear regression curves were generated in Prism5. Data sets with a sample size of ≤6 were assumed to be normal distributed and were shown as mean ± standard error of the mean. Statistical comparison used unpaired Student's t-testing with Welch's correction or one-way ANOVA testing. All data sets with a sample size of $n > 6$ were assessed by Shapiro-Wilk testing for normal distribution. If at least one data set did not pass the test ($P < 0.5$), the data were shown as Whiskers' plots, with 25th/75th percentiles indicated as blue lines, median as a red line, and minimum and maximum values as whiskers ends. These data sets were compared using nonparametric two-tailed Mann–Whitney testing. All test results are either included in the figure legend or are summarized in the Source Data file. For simplicity, data in curves were shown as the mean ± standard error of the mean, even when distributed non-normally. Full statistical information, including median, 25th/75th percentile, minimum/maximum values are provided in the Source Data file. Non-linear regression used "log(inhibitor) vs. response—Variable slope (four parameters)" settings in Prism5 (GraphPad). $EC_{50}$ values were determined graphically from these curves and corrected for the molecular weight of the counter ions in the respective compound. All graphs were generated in Prism5 and modified in CorelDraw X6. Acquired images were adjusted in brightness and contrast and gamma, using MetaMorph.

**Quantitative analysis of living fungal cells**. All experiments involving MALCs used stock solutions of 0.5–100 mg ml⁻¹ compound, dissolved in methanol. All control values used the methanol volumes that corresponded to the highest compound concentration, corresponding to the highest methanol volume used. All measurements were taken from distinct samples.

*(a) Cell survival assay:* To assess cell death, cells of Z. tritici IPO323_eGFP-SSO1 were grown for 48 h in YG liquid medium, thence various concentrations of $C_{12}$-$G^+$ were added and incubated for up to 3 h. 0.2 µl of Live/Dead™ Fixable Red Dead Cell Stain Kit (Thermo Fisher Scientific) or 1 µl propidium Iodide (Sigma-Aldrich) was added to 100 µl of treated cell suspension, and incubated for 5 min on a SB2 Rotator (Bibby Scientific Limited, Stone, UK), at RT. Stained cells were placed onto a 2% (w v⁻¹) agar cushion and analysed by fluorescence microscopy, with 488 nm and 561 nm lasers set at 20% intensity and exposure time of 150 ms.

*(b) Plasma membrane potential detection:* The effect of $C_{12}$-$G^+$ on plasma membrane potential was tested using the voltage-sensitive fluorescent dye DiBAC₄(3) (Bis-(1,3-dibutylbarbituric acid) trimethine oxonol; Thermo Fisher Scientific). Z. tritici cells (strain IPO323_mCh-ZtSso1) were grown for 2 days in YG liquid medium, incubated with various concentrations of $C_{12}$-$G^+$ for 30 min, followed by addition of 1 µl DiBAC₄(3) to a final concentration 1 µg ml⁻¹ and

0.2 μl of Live/Dead™ Fixable Red Dead Cell Stain Kit. Cells were incubated for 5 min on a SB2 Rotator (Bibby Scientific Limited) at RT, washed twice with YG media and analysed by fluorescence microscope, at 488 nm with 5% laser intensity and at 561 nm with 40% laser intensity.

(c) *Mitochondrial membrane potential:* Mitochondrial membrane potential was visualized with tetramethylrhodamine methyl ester (TMRM; Thermo Fisher Scientific). Cells of *Z. tritici* (strain IPO323_eGFP-Sso1) and *U. maydis* (strain FB1GSso1) were grown in their respective media (see above). One microliter TMRM was added to 1 ml of fungal cell suspension and incubated in the dark at RT for 10 min, whilst rotating on a SB2 Rotator (Bibby Scientific). Subsequently, cells were placed onto a 2% (w v$^{-1}$) agar cushion and imaged using 561 nm laser, at 20% intensity, and an exposure of 150 ms. The cell outline was recorded by imaging the GFP labeled plasma membrane, with a 488 nm laser at 20% intensity and an exposure time of 150 ms. In case of Guy11 spores were pipetted onto hydrophobic glass coverslips (Menzel-Glässer cover glasses, Thermo Fisher Scientific) and incubated in a wet chamber for 30 min at 25 °C. The cover glasses, with attached conidia, were floated upside down on 10 ml sterile water with different concentrations of $C_{12}$-G$^+$, $C_{18}$-SMe$_2^+$, and $C_{18}$-NMe$_3^+$ for 30 min, before staining with TMRM.

(d) *ATP detection in cytoplasmic extracts:* To monitor the effect of MALCs on ATP production, cells of wildtype *Z. tritici* strain IPO323 were grown for 48 h at 18 °C, at 200 rpm, in YG liquid medium. 10–20 ml of cell suspension were treated for 1–4 h ($C_{12}$-G$^+$) or 2 h ($C_{18}$-SMe$_2^+$ or $C_{18}$-NMe$_3^+$) at 5 μg ml$^{-1}$, at which mitochondria were largely depolarized (see Fig. 4g; Supplementary Fig. 4). Control experiments were incubated with methanol. Three milliliter treated cell suspension was centrifuged 5000 rpm for 5 min in a Micro Star 17R cooled centrifuge (VWR, Lutterworth, UK). The sedimented cells were resuspended in 100 μl of 10 mM Tris-HCL buffer (pH 7.0). Acid-washed glass beads (425–600 μm bead size; Sigma-Aldrich) were added and the cells ruptured in a IKA Vibrax shaker (IKA, Staufen, Germany). The mixture was centrifuged (10 min at 11,200 rpm, 4 °C) and ATP was detected in 5 μl of the supernatant, using the luciferase-based ATP Determination Kit (A22066; Thermo Fisher Scientific) as per suppliers' instructions. Luminescence was detected via a GloMax® Discover plate reader (Promega, Wisconsin, USA).

(e) *Detection of dissolved oxygen:* *Z. tritici* strain IPO323 was grown in fully-transparent minimal medium[69] for 48 h at 18 °C, under oxygen saturation conditions. Twenty milliliter samples of cell suspension (optical density of OD$_{600}$ = 0.4) was treated with 5 μg ml$^{-1}$ (the concentration used to determine cellular ATP levels); methanol was used in control experiments. After 30 min of incubation, 15 ml of that treated culture was placed in a sealable reaction tube and incubated for 3 h at room temperature. The total amount of dissolved oxygen in these samples at $T = 0$ h and $T = 3$ h was analysed using the HI-3810 Dissolved Oxygen test kit (Hanna Instruments Ltd, Leighton Buzzard, UK), following the manufacturers' instructions.

(f) *Test for the activity of NADH oxidases in isolated mitochondria:* The NADH oxidase activity of complex 1 was investigated as described[38]. In brief, IPO323 cell extracts were generated from 200 ml overnight cultures, grown to OD$_{600}$ = 0.5–0.8. Cells were harvested by centrifugation at 5000 rpm for 10 min and the cell sediment was resuspended in 1 ml ice cold buffer A (0.6 M mannitol, 1 mM EDTA, 0.5% bovine serum albumin, 1 mM phenylmethylsulfonyl fluoride, 10 mM Tris-HCL, pH 7.5) and frozen in liquid nitrogen. The cells were disrupted in a cooled mixer mill (MM400; Retsch, Haan, Germany) at 30 stakes per second for 2 × 2.5 min and the powder resuspended in 2 ml buffer A. Cell debris was removed by 2-times low-speed centrifugation (1000 × g for 10 min, 4 °C) in a Heraeus Biofuge Stratos benchtop centrifuge (#3331 rotor, Heraeus, Hanau, Germany). Supernatants were centrifuged at 10,500 × g for 10 min, followed by two washing steps, using centrifugation and 5 ml of ice-cold buffer B each (0.6 M mannitol, 1 mM EDTA, 1% BSA, 10 mM Tris-HCl, pH 7.0). Mitochondria were resuspended in 200–400 μl of cold buffer C (0.6 M mannitol, 10 mM Tris-HCl, [pH 7.0]), and the protein content determined using a Qubit™ Protein Assay Kit (Q33212, Thermo Fisher Scientific) and a Qubit® 2.0 Fluorometer (Thermo Fisher Scientific). These mitochondrial preparations were retained on ice for subsequent usage. We tested if this purification protocol yields "healthy" mitochondria, by purifying the organelles from strain IPO323_Acd1-ZtGFP, followed by staining with TMRM. Co-imaging of red- and green-fluorescence was described above.

To analyse NADH oxidase activity, 20–25 μg of mitochondrial protein in buffer C was added to 0.8 ml H$_2$O, supplemented with the 5 μg ml$^{-1}$ $C_{12}$-G$^+$, $C_{18}$-SMe$_2^+$, or $C_{18}$-NMe$_3^+$. Control experiments were incubated with methanol (positive control) or 100 μM rotenone/50 μM diphenyleneiodonium (negative control). The mixtures were pre-incubated for 15 min at room temperature, followed by addition of 0.2 ml reaction mixture (5 mg ml$^{-1}$ bovine serum albumin, 0.24 mM KCN, and 0.8 mM NADH, 50 μM 2,3-dimethoxy-5- methyl-6-n-decyl-1,4 benzoquinone. NADH+ absorbance at 340 nm, corrected for the baseline at 380 nm, was measured after 1 h incubation in the dark at room temperature, using a Jenway 6705 UV/visible spectrophotometer (Jenway®, Stone, UK).

(g) *Detection of reactive oxygen species:* ROS was detected by staining with Dihydrorhodamine 123 (DHR123, Sigma-Aldrich). Cells of *Z. tritici* (IPO323_mCh-ZtSsoI), *U. maydis* (strain FB1) were grown in their respective media (see above), and incubated with 5 μg ml$^{-1}$ $C_{12}$-G$^+$, $C_{18}$-SMe$_2^+$, and $C_{18}$-NMe$_3^+$ or 5 and 20 μg ml$^{-1}$ $C_{12}$-DMS$^+$ and $C_{16}$-SMe$_2^+$, for 30 min or 24 h at RT. DHR123 was added to the cell suspensions to a final concentration 2 μg ml$^{-1}$, and

cells were incubated for 15 min at RT, in the dark, on a SB2 Rotator (Bibby Scientific). The cells were placed on a 2% (w v$^{-1}$) agar cushion and DHR123 fluorescence recorded using the described microscopic set-up (488 nm laser at 10% intensity, exposure time of 150 ms). With *Z. tritici* a reference image was taken using a 561 nm laser at 20%, whereas a bright-filed image was taken for *U. maydis* cells. *M.oryzae* Guy11 spores were pipetted onto hydrophobic glass coverslips (Menzel-Glässer, Thermo Fisher Scientific, Loughborough, UK), as described in (d), with different concentrations of $C_{12}$-G$^+$, $C_{18}$-SMe$_2^+$ and $C_{18}$-NMe$_3^+$ for 30 min before staining with DHR123. The DHR123 fluorescence was recorded using the 488 nm laser at 10%. Fluorescent intensities were measured using MetaMorph (Molecular Devices) and values were corrected for the image background.

(h) *Inhibiting the fungal respiration chain and mROS development:* ROS levels were altered by using 100 μM rotenone and 10 μM antimycin A, obtained from Sigma-Aldrich. *Z. tritici* (IPO323_mCh-ZtSso1) cells were grown in YG liquid medium for 2 days at 18 °C followed by a 30 min or 24 h incubation with 100 μM rotenone, 10 μM antimycin A, 10 μM antimycin A + $C_{12}$-G$^+$ or $C_{18}$-NMe$_3^+$ or 100 μM rotenone + $C_{18}$-SMe$_2^+$. DHR123 was added to cell suspensions, to a final concentration 2 μg ml$^{-1}$, and cells were incubated for 15 min at RT, in the dark, on a SB2 Rotator (Bibby Scientific). The cells were placed on a 2% (w v$^{-1}$) agar cushion and DHR123 fluorescence was recorded using the 488 nm laser at 10% intensity and an exposure time of 150 ms.

(i) *Detection of apoptotic cells via caspase activity:* Caspase activity was monitored using the CaspACE FITC-VAD-fmk In Situ Marker assay (Promega). *Z. tritici* cells, strain IPO323_mCh-ZtSsoI, grown in YG liquid medium was treated with 5 μg ml$^{-1}$ $C_{12}$-G$^+$, $C_{18}$-SMe$_2^+$, and $C_{18}$-NMe$_3^+$ for 24 h. Hundred microliter cell suspension was supplemented with 0.1 μl of CaspACE FITC-VAD-fmk and 0.1 μl propidium iodide and incubated for 10 min at RT in the dark on a SB2 Rotator (Bibby Scientific Ltd.). Cells were sedimented and resuspended in 100 μl of fresh media and placed onto a 2% (w v$^{-1}$) agar cushion. Early apoptotic cells were identified by green staining with CaspACE FITC-VAD-fmk, but no staining with propidium iodide. Microscopic observation was done as described above, using 488 and 561 nm lasers at 40 and 80% intensity, respectively, and an exposure time of 150 ms.

(j) *Detection of apoptotic cells with annexin V staining:* Phosphatidylserine exposure in apoptotic cells was detected using the Annexin-V-FLUOS staining Kit (Roche, Basel, Switzerland), following the manufacturer's protocol. *Z. tritici* strain IPO323_mCh-ZtSsoI cells were grown in YG liquid medium and treated for 24 h with 5 μg ml$^{-1}$ $C_{12}$-G$^+$, $C_{18}$-SMe$_2^+$, and $C_{18}$-NMe$_3^+$. Cells were centrifuged at 6000 rpm for 5 min and re-suspended in 100 μl of 2% Annexin-V-Fluorescein and 2% Propidium iodide in incubation buffer. After 10 min incubation at RT in the dark on a SB2 Rotator (Bibby Scientific Ltd.), the cells were pelleted and washed with fresh media. Cells were deemed apoptotic when stained with the green-fluorescent Annexin-V dye, but not with red-fluorescent propidium iodide.

(k) *Germination inhibition assay:* The effect of $C_{12}$-G$^+$, $C_{18}$-SMe$_2^+$, and $C_{18}$-NMe$_3^+$ on germination of *M. oryzae* Guy11 conidia was performed as described in (d) with hydrophobic glass cover, with attached conidia, which were floated upside down on sterile water with different concentrations of $C_{12}$-G$^+$, $C_{18}$-SMe$_2^+$, and $C_{18}$-NMe$_3^+$ for 3 h. DIC images were taken and the percentage of germinated conidia recorded in MetaMorph (Molecular Devices).

**Plant-related experiments.** Wheat, *Triticum aestivum* cultivar Galaxy, was grown for 14 days in a Fitotron SGC 120 growth chamber (Weiss Technik UK, Loughborough, UK), 14 h light (intensity 500 μmol of PAR), 24 °C, 80% RH (relative humidity); 10 h dark, 20 °C, 60% RH with automated watering. Rice, *Oryza sativa*, cultivar CO-39 was grown in a greenhouse for 21 days 16 h light (200–500 μmol m$^2$ s)/8 h dark cycle at 23–35 °C and RH of 50–70%.

(a) *Disease control assays:* To test the ability of MALCs to protect wheat against Septoria tritici leaf blotch, the lower surface of the second leaf of ten individual 14-day-old wheat plants was sprayed with 1 ml of various concentrations of $C_{12}$-G$^+$, $C_{18}$-SMe$_2^+$, or $C_{18}$-NMe$_3^+$ (stocks 25 mg ml$^{-1}$ in methanol) and 0.04% (v v$^{-1}$) Tween 20 (Sigma-Aldrich), using a Voilamart AS18 airbrush/compressor (Hydirect, Guang Dong, China) set at 5 psi. The control solution contained equivalent volumes of the solvent methanol. After 24 h growth, plants were spray-inoculated at 5 psi with 1 ml of cell suspension of *Z. tritici* (strain IPO323; 3 × 10$^5$ cells per ml). Pathogen cell numbers were determined using a Cellometer Auto 1000 cell counter (Nexcelom Biosciences, Lawrence, USA). Plants were grown under clear plastic bags for 3 days, followed by additional 15 days in the growth cabinet. Leaves were harvested and incubated in wet-chambers 25 °C for 3 days to ensure pycnidia development. At 21 dpi detached leaves were scanned, using an Epson Perfection V850 Pro scanner (Epson, Hemel Hempstead, UK). The area covered by pycnidia was determined using software ImageJ (https://imagej.net/Downloads) and compared to total leaf area.

To test whether MALCs protect rice plants against rice blast disease, 21-day-old rice leaves (cultivar C0-39) were detached, laid abaxial surface uppermost on 0.5% (w w$^{-1}$) water agar plates and sprayed with various concentrations of $C_{12}$-G$^+$, $C_{18}$-SMe$_2^+$, or $C_{18}$-NMe$_3^+$. Mixtures contained 0.2% (w v$^{-1}$) gelatin. Treated rice leaves were incubated for 24 h at RT, followed by spray-inoculation with 1 ml of *M. oryzae* spore suspension (strain Guy11, 1 × 10$^5$ cells ml$^{-1}$). After 4 days of

incubation at RT and daylight, leaves were scanned and lesion formation was determined in ImageJ.

*(b) Detection of hydrogen peroxide production during an oxidative burst:* Twenty one-days-old leaves of rice cultivar CO-39 (8–10 leaves per treatment) were sprayed with 2 ml solution containing 100 μg ml$^{-1}$ $C_{12}$-$G^+$, $C_{18}$-$SMe_2^+$, and $C_{18}$-$NMe_3^+$ and 0.2% (w v$^{-1}$) gelatin as described above. The negative control contained 0.4% of the solvent methanol, the positive control contained 15 mM salicylic acid (Sigma-Aldrich), a known inducer of hydrogen peroxide production in plants. Plants were incubated for 6 h, the second leaf detached and submerged in 1 mg ml$^{-1}$ 3,3-diaminobenzidine in deionised water, supplemented with 0.04% (w v$^{-1}$) Tween20, and incubated for 24 h at RT. Subsequently, leaves were washed in deionised water for 30 min and immersed in 100% ethanol for 3 days at RT. Cleared leaves were scanned, using an Epson Perfection V850 Pro scanner; DAB-stained areas were determined using the "Hue" and "Brightness" functions in the "Color Threshold" tab in ImageJ. The potential of each MALC to induce hydrogen peroxide production was estimated as a percentage DAB-stained of leaf area.

*(c) Determining plant biomass after $C_{18}$-$SMe_2^+$ treatment:* To investigate if $C_{18}$-$SMe_2^+$ affects plant growth, 14-day-old rice seedlings (7–10 per pot) were spray-treated with 2 ml of 100 or 150 μg ml$^{-1}$ $C_{18}$-$SMe_2^+$ in 0.2% (w v$^{-1}$) gelatin. Control plants were sprayed with water containing an equal amount of the solvent methanol and gelatin, as above. Greenhouse grown plants were resprayed on day 28 as per the first treatment. After 14 days further growth, all aerial plant parts were harvested and the wet weight determined.

**Toxicity testing.** *(a) Phytotoxicity test:* One milliliter test sample solution was prepared by adding $C_{12}$-$G^+$, $C_{18}$-$SMe_2^+$, and $C_{18}$-$NMe_3^+$ (stocks 25 mg ml$^{-1}$ in methanol) at 1 mg ml$^{-1}$ and 0.04% (v v$^{-1}$) Tween 20 (Sigma-Aldrich) to ddH₂O. The control solution contained equivalent concentrations of methanol and Tween. Wheat and rice leaves were sprayed as described above. Wheat plants were incubated in the Fitotron SGC 120 growth chamber, rice leaves were incubated in wet chambers at RT for 7 days. The apical 10 cm of leaves were scanned and a change in color was assessed in areas at 1–4 cm behind the tip of the leaf, using ImageJ. A color thresh-hold with "Hue" range of 50–100 was considered "green", yellow areas had ranged at 40–50 and brown covered a range of 0–40. The percentage of leaf area per color was calculated and plotted using Prism7 (GraphPad Software). Green tissue was deemed healthy and yellow/brown recorded as an index of leaf phytotoxicity.

*(b) Test for effect of MALCs on mitochondria in human cultured cells:* Human skin fibroblasts (C109) were cultured in DMEM, high glucose (4.5 g l$^{-1}$; Thermo Fisher Scientific), supplemented with 10% (v v$^{-1}$) FBS, 100 U ml$^{-1}$ penicillin, and 100 μg ml$^{-1}$ streptomycin, at 37 °C, with 5% CO₂ and 95% humidity. Fragmentation of mitochondria was determined by immunofluorescence microscopy. To this end, cells (2.5 × 10⁵) were seeded onto glass coverslips in DMEM and incubated for 24 h at 37 °C, 5% CO₂ prior to treatment. After 30 min of treatment with MALCs, cells were washed with phosphate buffered saline (PBS, pH 7.4) and fixed with 4% (w v$^{-1}$) paraformaldehyde in PBS, pH 7.4. Subsequently, cells were permeabilized with 0.2% (v v$^{-1}$) Triton X-100, treated with 1% (w v$^{-1}$) BSA, and incubated with anti-ATPB (mc mouse, dilution 1:3000; ab5432, Abcam, Cambridge, UK) and secondary antibodies (Alexa488 anti-mouse IgG(H+L), dilution: 1:400; Molecular Probes/Life Technologies). Images were taken as described above. For live-cell imaging, 10⁵ cells were grown on 3.5-cm glass bottom dishes (Cellview; Greiner BioOne, Germany) for 24 h at 37 °C, 5% CO₂. MALCs were applied in fresh DMEM and incubated for 30 min. One microliter of Image-iT™ TMRM (Thermo Fisher Scientific) was added in the dark, and cells were incubated for 5 min before washing twice with PBS and 2 ml of pre-warmed HEPES-buffered DMEM, high glucose (4.5 g l$^{-1}$), without phenol red (Thermo Fisher Scientific) containing 10% (v v$^{-1}$) FBS. Imaging was done as described above using a controlled-temperature chamber and an objective warmer at 37 °C.

*(c) MTT assay:* Human skin fibroblasts (C109) and human hepatoblastoma cells (HepG2) were seeded into 96-well culture plates (Greiner bio-one, Austria) at 5 × 10³–1 × 10⁴ cells in each well in 200 μl culture medium (10% FBS, Gibco Life Technologies, USA) and incubated for 24 h (37 °C, 5% CO₂, 95% humidity). Cells were washed with phosphate buffered saline, pH7.4, and 100 μl medium alone (blank), supplemented with MALCs (0.5–10 μg ml$^{-1}$, 8 replicas) or the solvent methanol (8 replicas) was added. Cells were incubated for 24 h, followed by the addition of 100 μl of culture medium per well, containing 500 μg ml$^{-1}$ 3-(4,5-dimethylthiazol-2-yl)-2,5-diphenyltetrazolium bromide (MTT; Sigma Aldrich/Merck, Darmstadt, Germany). This was followed by 3 h incubation as above. The supernatant in each well was discarded, and precipitated formazan crystals were dissolved by addition of 150 μl DMSO (Sigma Aldrich/Merck, Germany). The absorbance at 560 nm was measured using the GloMax® Discover multimode plate reader (Promega). Measurements were corrected for the blank value. All data were compared to the solvent control, which was set to 100%.

*(d) Daphnia magna toxicity test:* Ten microliter of water containing ~20 *Daphnia magna* were exposed to different concentrations of $C_{12}$-$G^+$, $C_{18}$-$SMe_2^+$, and $C_{18}$-$NMe_3^+$ or, in case of control samples, 0.025% (v v$^{-1}$) of the solvent methanol. The zooplankton-containing mixtures were transferred to a 50 ml tube and incubated at gentle shaking for 24 h. Living water fleas were identified in petri

dishes. These were either motile, showed escape behavior when disturbed or showed internal motility (e.g., gill movement). The change in mitochondrial membrane potential in treated water fleas was determined with Image-iT™ TMRM Reagent (Thermo Fisher Scientific). To this end, 1 ml samples, containing *Daphnia magna*, were supplemented with 1 μg ml$^{-1}$ of $C_{12}$-$G^+$, $C_{18}$-$SMe_2^+$, and $C_{18}$-$NMe_3^+$ for 30 min, and 1 μl TMRM stock solution. This mixture was incubated for 10 min on a SB2 Rotator (Bibby Scientific Limited). For microscopic observation, water fleas were immobilized using Protoslow ACS 020 (Blades Biological Ltd., Edenbridge, UK) and directly imaged using a 4× UPlanFLN 0.13 objective (Olympus).

*(e) AMES test for genotoxicity:* The AMES plate incorporation experiment was performed according to OECD guideline 471 by Gentronix Ltd., Cheshire, UK, following published procedures[55,56]. In brief, various concentrations of $C_{18}$-$SMe_2^+$ (0.05, 0.16, 0.5, 1.6, 5, 16, 35, and 50 μg per plate) were tested for their mutagenic potential in *Salmonella typhimurium* histidine-requiring LT2 strains TA1535 (*hisG46*, base pair substitution), TA1537 (*hisC3076*, frameshift), TA98 (*hisD3052*, frameshift, plasmid pKM101), TA100 (*hisG46*, base pair substitution, plasmid pKM101) and *Escherichia coli* tryptophan-requiring WP2 strain *uvrA*/pKM101 (*trpE*, base pair substitution, plasmid pKM101), both in the presence (+S9, metabolic activation) and absence (−S9) of a post-mitochondrial fraction of rat liver homogenate. A hundred microliter of freshly grown over-night cultures were mixed with equal volumes of the solvent dimethyl sulfoxide (DMSO, negative control), mutagenic agents (positive control; for +S9 experiments: 2-aminoanthracene for all samples; for −S9 experiments: sodium azide for TA1535 and TA100, 9-aminoacridine.HCl for TA1537, 2-nitrofluorene for TA98, potassium dichromate for *uvrA*/pKM101), or test item formulation dilution. Subsequently, 500 μl buffer (−S9) or S9 fraction (+S9) was added. Each sample was mixed with 2 ml 0.6% (w v$^{-1}$) "soft top-agar", supplemented with low amounts of histidine- or tryptophan and poured onto minimal medium glucose agar plates, with 3 replicas per sample. The plates were incubated for 3 days at 37 °C and mutant colonies, reverted from auxotrophic to prototrophic behavior, were counted and counts normalized to the solvent control count.

**Synthetic chemistries.** $C_{12}$-$NMe_3^+$, $C_6$-$NMe_3^+$, $C_{12}$-$PO_4^{2-}$, $C_{18}$-$NMe_3^+$, and $C_{12}$-$G^+$ were obtained from Sigma-Aldrich ($C_{12}$-$NMe_3^+$: #D8638, $C_6$-$NMe_3^+$: #53272, $C_{12}$-$PO_4^{2-}$: #CDS000572, $C_{12}$-$SO_4^-$: #1614363, $C_{18}$-$NMe_3^+$: #359246; $C_{12}$-$G^+$: #45466). Other MALCs used in this study were synthesized as follows:

1-Dodecyltriethylammonium bromide ($C_{12}$-$NEt_3^+$): 1-Bromododecane (4.46 cm³, 19.0 mmol) was added to a solution of triethylamine (2.11 cm³, 22.3 mmol) in butanone (5 cm³) and the stirred mixture was heated under reflux for 24 h. After cooling to room temperature, the buff-colored solid formed, was filtered off and washed with diethyl ether (3 × 30 cm³), before drying in vacuo. The title compound was obtained as a buff, crystalline solid (4.63 g, 70%); m.p. 143–144 °C; $\nu_{max}$ cm$^{-1}$ (ATR) 2975 (w), 2590 (m), 2914 (s), 2848 (s), 1494 (w), 1478 (m), 1466 (s), 1396 (m), 1374 (w), 1166 (w), 1009 (m), 814 (m), 800 (m) and 719 (m); $\delta_H$ (400 MHz; CDCl₃) 0.85 (3H, t, $J$ = 7.2 Hz, $CH_3CH_2CH_2$), 1.17–1.30 (18H, br m, $CH_3(CH_2)_9$), 1.36 (9H, t, $J$ = 7.2 Hz, ($CH_3CH_2$)₃N), 1.62–1.69 (2H, br m, $CH_2CH_2N$), 3.22–3.26 (2H, m, ($CH_2$)₉$CH_2N$) and 3.48 (6H, q, $J$ = 7.2 Hz, ($CH_3CH_2$)₃N); $\delta_C$ (100 MHz; CHCl₃) 7.97 (($CH_3CH_2$)₃N), 13.98 ($CH_3CH_2CH_2$), 21.93 ($CH_3CH_2CH_2$), 22.52, 26.32, 29.00, 29.15, 29.25, 29.29, 29.42, 31.73 ($CH_3CH_2(CH_2)_9$), 53.39 (($CH_3CH_2$)₃N), and 57.37 ($CH_2CH_2N$).

1-Dodecyldimethylsulfonium iodide ($C_{12}$-$SMe_2^+$): 1-Bromododecane (1.50 cm³, 6.25 mmol) was added to a stirred suspension of sodium methanethiolate (438 mg, 6.25 mmol) in acetone (11 cm³) and the reaction mixture was heated under reflux for 24 h. After filtering the cooled reaction mixture, the solvent was evaporated in vacuo to give a colorless oil. Iodomethane (5.0 cm³, 80.3 mmol) was added to a solution of the crude reaction product, assumed to be 1-dodecylmethyl sulfide (502 mg, 2.13 mmol), in diethyl ether (3 cm³) and the resulting solution was allowed to stand at room temperature for 48 h. Evaporation of the reaction mixture to dryness in vacuo, gave the title compound as a pale yellow, waxy solid (292 mg, 35% for the salt-forming step); m.p. 89–92 °C; $\nu_{max}$ cm$^{-1}$ (ATR) 2991 (w), 2917 (s), 2849 (s), 1425 (m), 1377 (w), 1321 (w), 1297 (w), 1275 (w), 1250 (w), 1110 (m), 1039 (m), 996 (w), 973 (w), and 720 (m); $\delta_H$ (400 MHz; CDCl₃) 1.82 (3H, t, $J$ = 6.9 Hz, $CH_3CH_2$), 1.16–1.35 (16H, complex, 8× $CH_2$), 1.40–1.48 (2H, m, $CH_2$), 1.69–1.80 (2H, m, $CH_2CH_2S$), 3.30 (6H, s, ($CH_3$)₂S) and 3.65–3.71 (2H, m, $CH_2S$); $\delta_c$ (100 MHz; CDCl₃) 14.09 ($CH_3CH_2$), 22.64 ($CH_2$), 24.21 ($CH_2$), 25.63 (($CH_3$)₂S), 28.26, 29.00, 29.28, 29.42, 29.54, 29.58, 29.61, 31.86 (8× $CH_2$), and 43.04 ($CH_2S$).

Dodecane-1,12-diylbis(dimethylsulfonium) iodide (($SMe_2^+$)₂-$C_{12}$): Sodium methanethiolate (214 mg, 3.0 mmol) was added to a stirred solution of 1,12-dibromododecane (500 mg, 1.5 mmol) in acetone (7 cm³) and the resulting mixture was heated under reflux for 18 h. The cooled reaction mixture was filtered and evaporated in vacuo, to give a colorless solid, containing a fine white solid. This residue was partitioned between water (10 cm³) and dichloromethane (10 cm³) and the separated organic phase was washed with water (10 cm³) and saturated aqueous sodium chloride solution (10 cm³) before drying (MgSO₄), filtration, and evaporation in vacuo, to give a colorless oil. Iodomethane (6.0 cm³, 96.0 mmol) was added to a solution of a portion of the crude reaction product, assumed to be 1,12-bis(methylthio)dodecane (388 mg, 1.4 mmol), in diethyl ether (3 cm³) and the

resulting solution was stirred at room temperature for 18 h. The reaction mixture was evaporated to dryness in vacuo, to give the title compound as a white solid (594 mg, 76% for the salt-forming step); m.p. 72–81 °C; $\nu_{max}$ cm$^{-1}$ (ATR) 2968 (w), 2918 (w), 2849 (s), 1467 (m), 1420 (m), 1331 (w), 1278 (w), 1218 (w), 1172 (w), 1112 (w), 1041 (m), 994 (m), 924 (w), and 722 (m); $\delta_H$ (400 MHz; D$_2$O) 1.16–1.30 (12H, complex 6× $CH_2$), 1.31–1.41 (4H, m, 2× $CH_2$), 1.66–1.75 (4H, m, 2× $CH_2CH_2S$), 2.77 (12H, s, 2× $(CH_3)_2S$) and 3.15–3.22 (4H, m, 2× $CH_2S$); $\delta_C$ (100 MHz; D$_2$O) 29.39 ($CH_2$), 24.35 ($CH_3$), 21.35, 27.93, 28.16, 28.31, 28.50 ($CH_2$), and 43.17 ($CH_2S$).

1-Hexyldimethylsulfonium iodide (C$_6$-SMe$_2$$^+$): 1-Bromohexane (700 μl, 5 mmol) was added to a stirred suspension of sodium methanethiolate (351 mg, 5 mmol) in acetone (10 cm$^3$) and the reaction mixture was heated under reflux for 7 h and stirred for a further 72 h at room temperature. After filtering the cooled reaction mixture, the solvent was evaporated in vacuo to give a colorless oil. Iodomethane (5.0 cm$^3$, 80.3 mmol) was added to a solution of the crude reaction product, assumed to be 1-hexylmethyl sulfide, in diethyl ether (3 cm$^3$) and the resulting solution was stirred at room temperature for 24 h. Evaporation of the reaction mixture to dryness in vacuo, gave a colorless residue, which was partitioned between water (30 cm$^3$) and dichloromethane (10 cm$^3$). The separated aqueous phase was extracted with dichloromethane (2 × 10 cm$^3$) and evaporated to dryness in vacuo, to give a pale yellow sirup, which was triturated with diethyl ether (3 × 20 cm$^3$). This gave the title compound as a yellow solid (635 mg, 45% over two steps); m.p. 80–84 °C; $\nu_{max}$ cm$^{-1}$ (ATR) 2955 (m), 2919 (s), 2850 (s), 1465 (s), 1420 (s), 1376 (w), 1333 (w), 1293 (w), 1248 (w), 1045 (s), 1008 (m), 924 (w) and 723 (s); $\delta_H$ (400 MHz; CDCl$_3$) 0.90 (3H, t, $J$ = 7.3 Hz, $CH_3CH_2$), 1.28–1.40 (4H, m, 2x $CH_2$), 1.47–1.56 (2H, m, $CH_2$), 1.77–1.87 (2H, m, $CH_2$), 3.38 (6H, s, $(CH_3)_2S$) and 3.74–3.80 (2H, m, $CH_2S$); $\delta_C$ (100 MHz; CHCl$_3$) 13.94 ($CH_3CH_2$), 22.31 ($CH_3CH_2$), 24.18 ($CH_2$), 25.63 ($(CH_3)_2S$), 27.91 ($CH_2$), 31.09 ($CH_2CH_2S$), and 42.91 ($CH_2S$).

1-Hexadecyldimethylsulfonium iodide (C$_{16}$-SMe$_2$$^+$): A solution of 1-bromohexadecane (1.53 cm$^3$, 5.0 mmol) in acetone (10 cm$^3$) was added to sodium methanethiolate (351 mg, 5.0 mmol) and the reaction mixture was heated under reflux for 5 h. After filtering the cooled reaction mixture, the solvent was evaporated in vacuo to give a low melting, white solid. Iodomethane (5.0 cm$^3$, 80.3 mmol) was added to a solution of a portion of the crude reaction product, assumed to be 1-hexadecylmethyl sulfide (506 mg, 1.86 mmol), in diethyl ether (3 cm$^3$) and the resulting solution was allowed to stand at room temperature for 48 h. The reaction mixture was evaporated in vacuo, to give a white solid, which was broken up in diethyl ether (20 cm$^3$) and filtered off, washing with further portions of diethyl ether (2 × 20 cm$^3$) to give the title compound as a white solid (648 mg, 84% for the salt-forming step); m.p. 89–92 °C; $\nu_{max}$ cm$^{-1}$ (ATR) 2917 (s), 2847 (s), 2786 (w), 1485 (w),1467 (w), 1437 (m), 1108 (m), 995 (w), 746 (w), 720 (m) and 687 (m); $\delta_H$ (400 MHz; CDCl$_3$) 0.81 (3H, t, $J$ = 7.5 Hz, $CH_3CH_2$), 1.14–1.48 (26H, complex, 13× $CH_2$), 1.69–1.80 (2H, m, $CH_2$), 3.30 (6H, s, $(CH_3)_2S$) and 3.66–3.72 (2H, m, $CH_2S$); $\delta_C$ (100 MHz; CDCl$_3$) 14.06 ($CH_3$), 22.62, 24.16 (2x $CH_2$), 25.55 ($(CH_3)_2S$), 28.22, 29.00, 29.20, 29.23, 29.29, 29.43, 29.54, 29.59, 29.62, 31.70, 31.85, 34.23 (12x $CH_2$), and 42.83 ($CH_2S$).

1-Octadecyldimethylsulfonium iodide (C$_{18}$-SMe$_2$$^+$): A solution of 1-bromooctadecane (1.43 g, 4.3 mmol) in acetone (15 cm$^3$) was added to sodium methanethiolate (300 mg, 4.3 mmol) and the reaction mixture was heated under reflux for 24 h. After filtering the cooled reaction mixture, the solvent was evaporated in vacuo to give a white solid. Iodomethane (3.0 cm$^3$, 48.2 mmol) was added to a solution of a portion of the crude reaction product, assumed to be 1-octadecylmethyl sulfide (500 mg, 1.66 mmol), in diethyl ether (2.5 cm$^3$) and the resulting solution was allowed to stand at room temperature for 96 h, during which time, it solidified to a white solid, which was dried in vacuo. The product was broken up in diethyl ether (20 cm$^3$) and filtered off, washing with further portions of diethyl ether (2 × 20 cm$^3$) to give the title compound as a white solid (440 mg, 90% for the salt-forming step); m.p. 81–97 °C; $\nu_{max}$ cm$^{-1}$ (ATR) 2966 (m), 2916 (s), 2849 (s), 1473 (m), 1465 (m), 1373 (w), 1332 (w), 1294 (w), 1235 (w), 1112 (w), 1069 (w), 1041 (m), 1008 (m), 988 (w), 924 (w), 723 (m) and 715 (m); $\delta_H$ (400 MHz; CDCl$_3$) 0.89 (3H, t, $J$ = 7.2 Hz, $CH_3CH_2$), 1.22–1.41 (28H, complex, 14× $CH_2$), 1.46–1.56 (2H, m, $CH_2$), 1.78–1.87 (2H, m, $CH_2CH_2S$), 3.38 (6H, s, $(CH_3)_2S$) and 3.72–3.78 (2H, m, $CH_2S$); $\delta_C$ (100 MHz; CDCl$_3$) 14.11 ($CH_3CH_2$), 22.67, 24.34 (2× $CH_2$), 25.72 ($(CH_3)_2S$), 28.32, 28.83, 29.00, 29.17, 29.24, 29.35, 29.45, 29.52, 29.59, 29.64, 29.68, 31.91, 34.30 (14× $CH_2$), and 43.53 ($CH_2S$).

All newly-synthesized compounds were characterized by $^1$H and $^{13}$C NMR spectroscopy and used at >98% purity.

**Estimating molecule lipophilicity**. To estimate the lipophilicity index[20] (LogP), we obtained structure data files (SDF files) from the ChEB1 server (https://www.ebi.ac.uk/chebi/) or the PubChem server (https://pubchem.ncbi.nlm.nih.gov/compound/) and translated these into line notations ("simplified molecular-input line-entry system", SMILES) using the "Online SMILES Translator and Structure File Generator" (https://cactus.nci.nih.gov/translate/). SMILES notations for C$_{18}$-SMe$_2$$^+$, MitoQ, Mito-Vitamin C, Mito-Metformin, and K20 were generated using ChemDraw Professional 15.0, PerkinElmer Ltd., Buckinghamshire, UK. SMILES notations were used to estimate LogP as an average of five prediction methods in

SwissADME (http://www.swissadme.ch[70]). All SMILES notations and LogP values used or discussed in this study are listed in Supplementary Table 1.

## Data availability
The authors confirm that all relevant data are included in the paper or in the Supplementary Information file. The source data underlying Figs. 1a–k, 2a–k, 3a–d, 4a–h, 5a–h, 6a–f, 7a–l, 8a–g and Supplementary Figs. 2, 3a, b, 4a, b, 5, 7a–e, 8a, b, 9 are provided as a Source Data file. Additional information is available from the authors upon request.

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

## Acknowledgements

This work was funded, in part, by BBSRC grants BB/I025956/1 and BB/P018335/1 awarded to principal investigator G.S. and co-investigator S.J.G. M.S. is funded by BBSRC grant BB/R016844/1. S.J.G. is a CIFAR Fellow "The Fungal Kingdom: Opportunities and Threats". The research described herein is covered by patent application GB 1904744.8. This patent shall be prosecuted in line with Exeter's ethical licencing pledge, which supports technology access to low and middle income countries. The authors wish to thank Dr. Christian Hacker, Bioimaging Center, University of Exeter, for performing electron microscopy and Dr. Christine Mee (Gentronix Ltd.) for performing the AMES test.

## Author contributions

G.S. conceived and coordinated the project, wrote the manuscript, prepared all figures and analysed data, including the molecule lipophilicity; M.Schu. performed all fungal live cell imaging experiments, did the *Daphnia magna* toxicity experiments and analysed microscopic data; S.J.G. discussed results and conceived experiments; T.A.S., M.Schu., and M.Schr. performed human cell culture experiments; M.W. synthesized MALCs and supported the lipophilicity analysis; S.K. generated strains; A.E. performed plant-related experiments. All authors contributed to the writing of the Methods section.

## Competing interests

G.S., S.J.G., and M.W. are listed as inventors in a related patent application (GB 1904744.8). The remaining authors declare no competing interests.
