## [Peer Review File · Nature Communications]

Reviewers' comments:

Reviewer #1 (Remarks to the Author):

The paper by Steinberg and co-workers describes the development and investigation of a novel non-toxic crop protectant fungicide. Significant amount of experimental data and intriguing discovery have been presented. The reported cationic surfactants appear to have promising biological activity. However, there are several issues or concern. Major revision is needed.

Major concern:

1. The reported cationic surfactants, C18-SMe₂⁺ or C12-G⁺ are simple in structure. Therefore, to make the claim of antifungal but low cytotoxic, convinced evidence must be carefully examined. From the reported data and literature precedents, it is obvious that cationic surfactants are antifungal. However, the methods/experiments of demonstrating low cytotoxicity is basing on the measurement or imaging of mitochondria fragmentation using only skin fibroblast. There is no information on how difference that cationic surfactants may affect the membranes of mammalian cells vs. fungal cells. In order to act on mitochondria, these cationic surfactants need to permeablize cell membranes: targets that should be investigated at least equally. While mammalian cell membrane is quite different from fungal membrane, is the mammalian mitochondria membrane also different from fungal mitochondria membrane? There is no explanation as why cationic surfactants is more active against the inner membrane of fungal mitochondria. Given the simplicity in the structure of cationic surfactants, it is difficult to conceive the difference in activity. The result from water flea experiment may partly support the claim of low cytotoxicity. Nevertheless, crustaceans are quite different form mammals. Perhaps, the authors should use MTT cell proliferation assay with more cell lines, and determine the IC₅₀.
2. The reported finding of cationic surfactants with multiple MOAs, protection against plant fungal pathogens, low phytotoxicity, and others may not be unprecedented. In fact, many crucial literatures may have been overlooked. There are ample reports (ex. *J. Antibiot.*, 2010, 63, 667; *Antimicrob. Agents Chemother.*, 2015, 59, 4861; *ACS Infect. Dis.*, 2018, 4, 825; *Front. Microbiol.*, 2014, 5, 671; *MedChemComm*, 2018, 9, 909) on the antifungal property of amphiphilic kanamycin, which is a type of cationic surfactant. Examples of antifungal amphiphilic kanamycin against plant and human pathogens while being low in cytotoxicity and phytotoxicity can be found, including even the "multiple MOA" (*Angew. Chem. Int. Ed.* 10.1002/anie.201809410) and "ROS production" (*ACS Infect. Diseases*, 2019, 5, 473). The SAR of "A longer alkyl chain improves anti-fungal activity" has also been demonstrated. Insights on why amphiphilic kanamycin is selective toward fungi at molecular basis have been reported. In short, the manuscript needs proper citation of literature and revision of some description.
3. The claims: "C18-NMe₃⁺ and C18-SMe₂⁺ show low toxicity in human cells and *Daphnia magna*" and "environmentally-benign" may be questionable. A similar antiseptic compound, cetrimonium bromide or hexadecyltrimethylammonium bromide, is noted for its toxicity and health concern. It is curious why C18-NMe₃⁺ and, perhaps, C18-SMe₂⁺ have low toxicity in human. In fact, from the chemistry perspective, sulfonium is a good leaving group making C18-SMe₂⁺ a potential alkylating agent. In addition, alkylsulfonium can yield sulfide ylide, which is reactive. Yet, there is no discussion on the stability of C18-SMe₂⁺ and the possible degraded products.

Minor issues:

1. No indication as what is the counter ions of the reported cationic surfactant.
2. Some experiments used only C12-G⁺ but not C18-SMe₂⁺, and vice versa. No explanation provided.

Reviewer #2 (Remarks to the Author):

Summary:

Steinberg et al, nicely summarize their efforts on deciphering the complex mode of action of a family of molecules referred as single alkyl chain cations (SACCs). The detailed characterization of the mode of action of dodine, a commercial SACC fungicide initially reported to act as a membrane disruptor was priming this research. The finer characterization of dodine mode of action using cell biology techniques paved the way to a more extensive characterization of different SACCs molecules which were generated in the frame of this work. An extensive range of cell biology and microscopic characterization assays were performed, mostly in *Z. tritici*, a major pathogen of wheat that is highly relevant for crop protection. This enabled the authors to systematically determine specific features in the mode of actions of this group of molecules. Among the SACCs molecules, a specific compound: C18-SMe₂⁺ displayed an interesting combination of effects involving the fungal mitochondria in which the molecule accumulates, inducing the formation of reactive oxygen species at the level of complex I and able to prime apoptotic cell death. The authors extended their analysis to two other relevant pathogens, *Magnaporthe grisea* (rice) and *Ustilago maydis* (corn) in which the SACCs molecules were shown to act in a similar way. Practical determination of in planta antifungal activity was determined and demonstrated the superior activity of two of the newly synthesized SACCs compared to dodine. Different assays, on human and fungal mitochondria, on daphnia and AMES tests enabled the authors to demonstrate encouraging levels of fungal specificity and therefore the potential for a safe toxicity profile in human and the environment. Initial resistance risk study enabled the authors to comment on a lower resistance risk for the SACCs compared to a single site SDHI (succinate dehydrogenase inhibitor) fungicide. The reported chemistry is very simple and despite the levels of activity were better than dodine the new SACCs may not translate into applied / commercial solutions, but this study clearly showcase the strategic relevance of in depth cell biology explorations to support mode of action elucidation and chemical design in the field of multisite fungicides. This study is exceptional and will therefore trigger broad attention from academics and the industry. It is a perfectly written report, containing appropriate literature references and in which hypothesis and conclusions are nicely supported by high quality data. Besides some reserves on the resistance risk assay which I think will require repetition/confirmation (Fig 7 g and h) or to be removed from the manuscript and few comments / amendments listed below I am happy to recommend this paper for publication in Nature Communications.

P2 L54 Change sentence to « This is well illustrated by the recent ban of Chlorothalonil by the European Union, a thiol-reactive fungicide pivotal for disease control, but with reported toxicity to aquatic organisms (ref) and bumblebees (ref).»

P3 L95 FRAC code list 2019 still reports dodine as "unknown MoA" -> change to 2019, insert link to reference website (www.frac.info).

P4 L105 Monitoring this MoA, we tested...

P4 L118 50% EC₅₀ -> remove 50%, add ± SD values obtained from biological replicates.

Fig1A EC₅₀ should be calculated using non-linear regression (four parameters) non-linear curve fitting using the Log values of AI concentration (easy to perform in Graphpad, which is the software you've reported using, transform x values to Log, then perform curve fitting). Maximum and minimum growth values/parameters can be fixed depending on visual examination of the fitted S-shaped curve. Represent the sensitivity curves using Log [AI] on the x axis. This comment applies to other figures: Figure 2h and j, Figure 7a, b, d ; Supplemental Figure 7a. EC₅₀ values should be expressed with ± SD.

P4 L119, not clear why capital letters were used for live / dead staining. Maybe the authors could summarize in one sentence the principle of this assay to clarify for the reader.

P4 L127 "This shorter treatment time and lower doses promised to....

P5 L152 Suggestion to join the 2 sentences: at concentrations that are 40 times higher to its EC₅₀ which strongly suggests that this plasma membrane effect is not the primary MoA of...

P5 L158 and found/observed that low concentration of.. -> an EC₅₀ of this effect should be calculated, it seems to be around 5ppm.

P6L171 Suggestion - at doses below 0.5ug/ml suggesting that IMM depolarization is the primary

MoA of

Table 1: Indicate values \pm SD and number of replicates / assays in the legend

Figure 3e: state concentration (at 10 μ g/ml) on the figure

P7 L227. Main sites of mROS formation are...

P8 L254. According to some cited literature (Kulkarni et al., 2019) it seems that the involvement of metacaspases in fungal PCD is a matter of debate, could the authors comment on that? The observation of caspase activation is visible but is that the hallmark of PCD or something else i.e. induced degradation of protein aggregates? The authors should develop on that / mitigate the message on PCD induction in the discussion.

P8 L258 concentration of SACCs / C18-SMe2+ -> actual concentration is not stated (likely 5 μ g.ml-1?), please do it in the text and on graphs 4g, 4i

P8 L262 It is implied by the similar values but are the same cells showing plasma membrane phosphatidylserine and caspase activity?

P8 L270 Our results show that depolarize the mitochondrial membrane of *Z. tritici* which should result in ATP synthesis inhibition. (actual effect on ATP is inferred, not shown)

P9 L278 -> Supplementary figure 7a -> respective EC50 could be displayed in the figure and / or mentioned in the text.

Figure 6c and d -> why not showing the Negative / uninfected controls? This is particularly relevant for wheat, is the 21DPI mock leaf still green or brown?

P10 L305 -> correct Supplementary figure number 98b -> 8b

P12 L375 -> make a sentence, displayed bactericidal activity at doses over ...

P12 Resistance section: The results with Fluxapyroxad are surprising, resistance development frequencies in the absence of mutagen are 100x higher than the ones reported for other SDHIs in the presence of a mutagenic treatment (frequency of $<1.10^{-6}$) (ref 56). Then the exclusive presence of the SDHB (H267L) mutation in the 11 resistant colonies is suspicious (multiple mutations are possible, check pubmed PMC5680630). Then one can see tiny colonies developing from WT spotting on 50x EC50 Fluxapyroxad in figure 7g. Either fluxapyroxad is highly mutagenic for *Z. tritici* or there is a problem somewhere. I think that the outcome and conclusion of a lower resistance risk for the SACCs molecules is right but it may be a good thing to repeat the experiment as a safety check. Is a contamination of the *Z. tritici* stock with the mutant possible? (This particular mutation has been used for targeted genomic integrations in other communications). I would recommend repeating the Fluxapyroxad experiment with the culture of a clone derived from single celled IPO323 in order to confirm this result. If the result (figure h) does not repeat in the same way I would remove this section, the inherent lower resistance risk of a multisite fungicide can be assumed with confidence in the discussion.

P12 L399 antifungals

P12 L400 environmental impact

P12 L401 Proposal: replace resilient here by "provide sustained control over the seasons that is ... should not be easily overcome by the emergence of resistance"

P13 L406 Proposal: such compounds had....

P15 L478 replace sentence.... The rapid development of resistance in market leader chemistries makes the identification of new fungicides a priority. More sustainable fungicides, combining lower risk of resistance development and a safe toxicity profile to humans and the environment are the current drivers in crop protection research.

P15 L480 New fungicides should..... target... to reduce the chances of resistance....

P15 L490 Against SACCs is inherently lower compared to...

P15 L491 Finally, we demonstrate...

P15 L492 ... *Daphnia magna* and that

P15 L495- 498, Prop: that section should go after L493 (multisite) and to adapt sentence: Furthermore, we showed that sulfonium head group in C18-SMe2+ combined multiple fungicidal mechanisms...

Methods section is well described, no comments

Reviewer #3 (Remarks to the Author):

Development of novel antifungals and improvements in the safety of current fungicides have had a great impact on public health and economics. The manuscript (MS) submitted by Steinberg et al. presents a comprehensive study of "[a] novel non-toxic crop protectant fungicide with multiple modes of action". The aforementioned fungicides are characterized as SACCs (single n-alkyl chain cations) resembling a cationic surfactant – dodecylguanidinium (C12-G+). Through the MoA study of C12-G+ on the wheat pathogen *Zymoseptoria tritici*, the authors confirm that the principal targets of C12-G+ are likely in the mitochondrial inner membrane and matrix, while noting lesser effects on the cytoplasm membrane. With regard to their mitochondrial morphology, membrane potential, mROS level and apoptosis activities, optimization of a group of SACCs shown in Figure 3a brings to the fore two candidates (C18-NMe3+ and C18-SMe2+). Both candidates have improved protective activity in wheat, corn and rice from *Z. tritici*, *Ustilago maydis* and *Magnaporthe oryzae* infection.

The MS is quite well written. However, from a scientific standpoint it seems to be more descriptive than productive in its search for the root mechanism of SACCs, even though it is clearly intended as such a mechanistic confirmation (cf. study of C12-G+ in Figure 1). The lack of classical mitochondrial function studies under SACCs is particularly disturbing, since this obscures the conclusion and severely limits informed speculation. The data is quite convincing insofar as it shows a link between the SACCs and mitochondrial electron transportation chain, but this link must be rigorously tested. The current morphologic changes in mitochondria, membrane potential and mROS are not enough to assume one has found the 'real' target of SACCs, since those changes can be simply the consequences of distressed mitochondria, and may have a variety of causes. Finally, the dye for measurement of mitochondrial membrane potential is TMRM, which is also a lipophilic cation, as SACCs mentioned in this study. The reduction of TMRM signal under SACCs treatment might be no more than the result of a competition between the two compounds.

I recommend that the authors address the following points before further consideration:

1) Please clarify the source of the drugs and solvent for stock preparation and working solutions. The control solvents in fluorescent and electron microscopy are mostly labeled as methanol. However, in at least one instance the solvent seems to be DMSO (page 21 line 674).

2) There is no doubt C18-SMe2+ reduces the toxicity of C12-G+ for humans and other organisms. However, the working concentrations of the drugs and the treatment/exposure durations are not consistent, making it difficult to easily compare the effects. For example, when comparing the safety issues of SACCs on human cell and *Daphnia magna*, the in vivo protective effects of C18-SMe2+ on wheat and rice in Figure 6c and Figure 6d are 50 ug/mL and 125 ug/mL, respectively, while the H2O2 induction in rice (Figure 6i) requires 100 ug/mL C18-SMe2+ for initiation of plant defense. At this concentration, the mitochondria in human cells (Figures 2g and 2i) have already started fragmentation and have begun to undergo the loss of mitochondrial structure. Furthermore, the testing range of the same drug for *Daphnia* toxicity is from 1 µg/mL to 5 ug/mL in Figures 7C & 7D. The safety of this drug for aquatic crustaceans needs to be further discussed, and some explanation needs to be offered for the choice of concentrations used in this assessment.

3) As the results suggest, C18-SMe2+ induces mROS production at respiration complex I (Figure 4C), which is supported by the observation that rotenone can turn off the disturbance of C18-SMe2+ on CI electron or proton transportation (whereupon the generation of mROS is then reduced). However, there is another possibility – that the insertion of C18-SMe2+ on inner membrane of mitochondria perturbs the structure of several protein complexes, including CI. As CI

is the most extensive machinery complex in ETC, it would not be surprising to see a greater effect of SACCs on CI than on CIII in this study. To rule out this possibility, an assay of the complex assembly and enzymatic activities for CI should be carried out by blue native PAGE and enzymatic assay in isolated mitochondria.

Reply to reviewer's comments

(reviewer statements are given in italics, replies are in Roman script)

Reviewer #1

We wish to thank the referee for his/her very constructive criticism. In addressing the comments, we were able to improve the manuscript significantly. In the following, we summarise our detailed reply.

The paper by Steinberg and co-workers describes the development and investigation of a novel non-toxic crop protectant fungicide. Significant amount of experimental data and intriguing discovery have been presented. The reported cationic surfactants appear to have promising biological activity. However, there are several issues or concern. Major revision is needed.

Major concern:

1. The reported cationic surfactants, C18-SMe₂⁺ or C12-G⁺ are simple in structure. Therefore, to make the claim of antifungal but low cytotoxic, convinced evidence must be carefully examined. From the reported data and literature precedents, it is obvious that cationic surfactants are antifungal. However, the methods/experiments of demonstrating low cytotoxicity is basing on the measurement or imaging of mitochondria fragmentation using only skin fibroblast.

Reply:

We thank the referee for this comment. Indeed, in the first version of this manuscript, we claimed low cytotoxicity on the basis of TMRM staining of mitochondria, combined with electron microscopy investigation of ultrastructure. To make this point stronger, we have now performed MTT cell proliferation assays on C109 fibroblasts and on HepG2 cells (as requested by the referee, see below). Both assays show that C18-SMe₂⁺ and C18-NMe₃⁺ are less toxic than C12-G⁺.

Summary of action:

- 1) We moderated our claims and emphasise that we compare the effect of the new compounds on human cells and *Daphnia magna* with that of C12-G⁺, which is in use as Syllit or dodine in the field (page 12; page 14; page 17)
- 2) We have included the MTT results on C109 and HepG2 cells in the manuscript (page 12; Methods: page 26/27; Table 1; Table 2; Figure 8c; Supplementary Figure 9)

There is no information on how difference that cationic surfactants may affect the membranes of mammalian cells vs. fungal cells. In order to act on mitochondria, these cationic surfactants need to permeablize cell membranes: targets that should be investigated at least equally.

Reply part1:

With all due respect, we disagree with the statement of the referee that lipophilic cations “need to permeablize cell membranes”. In fact, all fungicides, and even lipophilic cationic dyes of mitochondria enter the cell by passive diffusion through the membrane. This has to do with their lipophilicity, indicated by the LogP value (a commonly accepted and widely used parameter). If the logP value is too high, molecules stay in the membrane (e.g. phospholipids, LogP ~11), if the logP value is negative, molecules are not lipophilic enough to pass the membrane (e.g. ATP or glucose, LogP values are -3.39 and -2.26, respectively).

Lipophilic cationic molecules that enter the mitochondria by passive diffusion include various therapeutics (e.g. Mito-Vitamin C) and dyes, commonly used to visualise mitochondria in living cells (e.g. Rhodamine 123). These molecules have a LogP value of 2.15 (Rhodamine 123) to 7.41 (MitoQ), which corresponds well with the LogP of C12-G⁺, C18-SMe₂⁺ and C18-NMe₃⁺ (2.26, 4.24, 5.96). Thus, it is most likely that these three lipophilic cations are diffusing through the plasma membrane without altering its integrity. We have determined the LogP values for various molecules using several methods and included a new Supplementary Table 1, summarising this information.

Summary of action, part1:

- 1) The concept of LogP and lipophilicity and the importance for membrane passage is discussed (page 3, 2nd paragraph; page 5; page 14)
- 2) LogP of various molecules (known to pass into the mitochondria, stay in the membrane or not pass through the membrane) as well as some known mitochondrial fungicides were estimated, as an average of 5 methods. The LogP values are summarised in Supplementary Table S1, the methodology to estimate LogP values is provided in the Methods (page 28)

Reply, part2:

In the previous version manuscript, we stained *Z. tritici* cells with the voltage-sensitive reporter DiBAC₄(3). From these experiments, we concluded that the C12-G⁺ does not increase the permeability of plasma membrane for ions. As ions are very small (e.g. radius potassium ion: 0.133 nm; Stein & Wilfred D., Transport and Diffusion across Cell Membranes, 1986 Academic Press Inc. Harcourt Brace Jovanovich, Publishers pp.128-129), we excluded that C12-G⁺ causes lethality by breaking up the outer membrane.

In this revised version, we have redone this analysis, and included C18-SMe₂⁺ and C18-NMe₃⁺. We also took into account, that the voltage-sensitive dye DiBAC₄(3) stains cells which are already dead. This we had not previously accounted for, which led to the conclusion that there was a slight effect of C12-G⁺ on plasma membrane integrity. In the new analysis, such “live/dead” dye-positive cells are excluded, which makes the analysis more accurate. The result is, again, that none of the three lipophilic cations affect the plasma membrane.

The effect of C12-G⁺ on the plasma membrane was our focus, to understand the primary mode of action of this compound. The effect on human culture cells is now extensively addressed in MTT assays (done in 2 cell lines). These results, and the extensive additional data, has been included in this revised manuscript. The effect of C12-G⁺ on the permeability of naked human culture cells does not add to the mode of action of this molecule in fungi. Moreover, the toxicity aspects are now much better covered by the MTT assays. Consequently, and for the sake of focus, we have decided not to extend the analysis of the effect of C12-G⁺ on the plasma membrane of human fibroblasts.

Summary of action, part 2:

- 1) We redid the DiBAC₄(3) staining of C12-G⁺ treated cells, now counterstaining with live/dead stain to exclude dead cells, which absorb the dye non-specifically. These results are included in Page 5; Figure 1i, 1j, 1k; Figure 4g; Table 1; Methods: page 20.
- 2) We tested plasma membrane depolarisation in *Z. tritici* cells, treated with C18-SMe₂⁺ and C18-NMe₃⁺ at 50-times the EC₅₀; no effect on the plasma membrane integrity was seen. These results are now included in Figure 4g; Table 1; page 8; Methods: page 20.

3) Toxicity of C12-G⁺, C18-SMe₂⁺, C18-NMe₃⁺ in human fibroblasts (C109) and liver cells (HepG2) was tested in MTT assays. These results are now included in the manuscript (page 12; Methods: page 26/27; Table 1; Table 2; Figure 8c; Supplementary Figure 9)

While mammalian cell membrane is quite different from fungal membrane, is the mammalian mitochondria membrane also different from fungal mitochondria membrane? There is no explanation as why cationic surfactants is more active against the inner membrane of fungal mitochondria.

Reply:

We apologise if we were unclear in this point. Several comments and references were already included in the previous manuscript that (a) aimed to explain why lipophilic cations insert into the inner mitochondrial membrane and (b) highlight differences between mammalian and fungal mitochondria. In this revised manuscript we rephrased the text to make these points clearer. Moreover, we now add experimental support that C12-G⁺, C18-SMe₂⁺ and C18-NMe₃⁺ inhibit NADH oxidation in isolated mitochondria. This step is performed at complex I, which carries fungal-specific subunits, but also by fungal-specific alternative NADH oxidases. Thus, it seems possible that the specificity of these lipophilic cations is related to inhibitory activity in the early steps of the respiration chain.

Summary of action:

- 1) We have changed the introduction to better explain that the matrix is negative, and thus captures the cationic head groups, followed by insertion of the lipophilic moiety into the inner mitochondrial membrane (page 3). This does not explain the specificity (see below point 3), but explains the effect on the inner mitochondrial membrane.
- 2) In this revised manuscript, we included NADH oxidase activity measurements in isolated mitochondria, treated with C12-G⁺, C18-SMe₂⁺ and C18-NMe₃⁺. We performed control experiments, showing that GFP-labelled mitochondria from *Z. tritici* are still polarised. Using these mitochondrial preparations, we show consistently that these lipophilic cations inhibit the generation of electrons at the beginning of the respiration chain. The data are included in the main text (page 6/7; Figure 2j, 2k; page 8; Figure 4f; Methods: page 21, upper page 22)
- 3) We discuss more distinctly that the specificity of C12-G⁺, C18-SMe₂⁺ and C18-NMe₃⁺ could be due to an effect on fungal-specific subunits of respiratory complex I and/or alternative fungal-specific NADH dehydrogenases (page 15; Figure Legend, page 48).

Given the simplicity in the structure of cationic surfactants, it is difficult to conceive the difference in activity.

Reply:

We agree! This is reflected in our statement vague in the discussion and highlights that there are several possibilities of how SACCs (now called mono-alkyl lipophilic cations= MALCs) could inhibit oxidative phosphorylation. However, small changes in these molecules have a significant impact on their activity. This is best illustrated by the fact that C18-SMe₂⁺ triggers ROS production (and as such apoptosis), while C16-SMe₂⁺ does not. Both molecules only differ by the length of their alkyl-chain, and yet they clearly have different activities. Moreover, C18-SMe₂⁺ and C18-NMe₃⁺ have the same alkyl-chain length, yet again different effects on ROS development. Thus, despite their simplicity these molecules are still complex enough to trigger different responses in living cells.

The result from water flea experiment may partly support the claim of low cytotoxicity. Nevertheless, crustaceans are quite different from mammals. Perhaps, the authors should use MTT cell proliferation assay with more cell lines, and determine the IC50.

Reply:

We thank the referee for this excellent suggestion! Indeed, MTT assays are the standard for cytotoxicity in mammalian culture cells. We performed this assay with C109 fibroblasts and human HepG2 cells and determined the EC50 values as suggested. In both cell types, C18-SMe2+ and C18-NMe3+ are less toxic than C12-G+. We have highlighted in the revised manuscript, that these MTT experiments (and the water flea experiments) aim to compare the new chemistries with an existing fungicide (C12-G+ is used in the field as Syllit or Dodine).

Summary of action:

- 1) We emphasise that the new compounds are less toxic than C12-G+, which is in use as Syllit or dodine in the field (page 12; page 13; of page 17)
- 2) Toxicity of C12-G+, C18-SMe2+, C18-NMe3+ in human fibroblasts (C109) and liver cells (HepG2) was tested in MTT assays. These results are now included in the manuscript (page 12; Methods: page 26/27; Table 1; Table 2; Figure 8c; Supplementary Figure 9)

2. The reported finding of cationic surfactants with multiple MOAs, protection against plant fungal pathogens, low phytotoxicity, and others may not be unprecedented. In fact, many crucial literatures may have been overlooked. There are ample reports (ex. J. Antibiot., 2010, 63, 667; Antimicrob. Agents Chemother., 2015, 59, 4861; ACS Infect. Dis., 2018, 4, 825; Front. Microbiol., 2014, 5, 671; MedChemComm, 2018, 9, 909) on the antifungal property of amphiphilic kanamycin, which is a type of cationic surfactant. Examples of antifungal amphiphilic kanamycin against plant and human pathogens while being low in cytotoxicity and phytotoxicity can be found, including even the "multiple MOA" (Angew. Chem. Int. Ed. 10.1002/anie.201809410) and "ROS production" (ACS Infect. Diseases, 2019, 5, 473). The SAR of "A longer alkyl chain improves anti-fungal activity" has also been demonstrated. Insights on why amphiphilic kanamycin is selective toward fungi at molecular basis have been reported. In short, the manuscript needs proper citation of literature and revision of some description.

Reply:

We apologise if we gave the impression that we overlooked or excluded published literature. We attempted to cite relevant existing literature (which was positively recognised by reviewer 2) and, indeed, were aware of amphiphilic kanamycin (and in particular K20, as being strongly anti-fungal; see Shrestha, S. K. *et al.* Antifungal amphiphilic aminoglycoside K20: bioactivities and mechanism of action. *Front. Microbiol.* 5, 671, 2014). Indeed, this molecule consists of a large cationic head group and a single alkyl chain. However, K20 is predicted as not lipophilic. In fact, its LogP value (an indicator of lipophilicity) is as low as that of ATP or glucose (LogP of amphiphilic kanamycin is -3.06, glucose is -2.26; see Supplementary Table1 for SMILES descriptor and LogP predictions). A positive LogP value is a prerequisite for passage through the membrane (Refsgaard, H. H. *et al.* *In silico* prediction of membrane permeability from calculated molecular parameters. *J. Med. Chem.* 48, 805-811; 2005). Thus, like glucose or ATP, K20 is unlikely to pass through the membrane, which corresponds to the reported mode of action at the plasma membrane (summarised in Subedi, Y. P., AlFindee, M. N., Takemoto, J. Y. & Chang, C. T. Antifungal amphiphilic kanamycins: new life for an old drug. *Medchemcomm.* 9, 909-919; 2018).

Based on these data, we concluded that amphiphilic kanamycin is not a lipophilic cation. As this study focusses on lipophilic cations and their activity inside the cell, extensive discussion of amphiphilic kanamycin is beyond the focus of this manuscript. However, to acknowledge the outstanding work on K20, we included a note and 2 references in the manuscript, highlighting K20 as an example that illustrates progress in fungicide development.

Summary of action:

- 1) The concept of LogP and lipophilicity and the importance for membrane passage is discussed (page 3, 2nd paragraph; page 5; page 14)
- 2) LogP of various molecules, known to pass into the mitochondria, stay in the membrane or do not pass the membrane, as well as some known mitochondrial inhibitory fungicides was estimated as an average of 5 methods. The LogP value for amphiphilic kanamycin (K20) was included. The LogP values are summarised in Supplementary Table S1, the methodology to estimate LogP values is provided in the Methods (page 28)
- 3) To better highlight that we are considering lipophilic cations in this manuscript, we renamed this group “mono-alkyl lipophilic cations” (= MALCs”) throughout the manuscript.
- 4) The lipophilicity of K20 (the best amphiphilic kanamycin) and its membrane passage is briefly discussed (page 16). Two references for further reading are provided (reference 62, 63).

3. The claims: “C18-NMe3+ and C18-SMe2+ show low toxicity in human cells and Daphnia magna” and “environmentally-benign” may be questionable. A similar antiseptic compound, cetrimonium bromide or hexadecyltrimethylammonium bromide, is noted for its toxicity and health concern.

Reply:

Following the referee’s suggestion, we have now addressed the cytotoxicity of C18-NMe3+ and C18-SMe2+ in MTT assays. In these assays and in the *Daphnia* experiments, both compounds are clearly less toxic than C12-G+, which is used in the field as dodine/Syllit. Thus, it is unlikely, that the new MALCs are more harmful than dodine. However, extensive toxicity tests need to be performed, but these are beyond the scope of this study.

Summary of action: We highlight now that our toxicity assays need to be extended (page 17).

It is curious why C18-NMe3+ and, perhaps, C18-SMe2+ have low toxicity in human. In fact, from the chemistry perspective, sulfonium is a good leaving group making C18-SMe2+ a potential alkylating agent. In addition, alkylsulfonium can yield sulfide ylide, which is reactive. Yet, there is no discussion on the stability of C18-SMe2+ and the possible degraded products.

Reply:

Comparisons of potential toxicity of C18-SMe2+ with the quaternary ammonium salts cetrimonium bromide and hexadecyltrimethylammonium bromide are, at best, tenuous. The chemistry and degradation pathways for the two different classes of compounds (and the possible toxicity of the compounds themselves and any degradation products) are likely to be very different. As recorded above, the cytotoxicity experiments go a long way towards addressing possible concerns. We recognised a potential alkylating activity of the sulfonium salt C18-SMe2+ and tested this in a professionally done AMES test according to OECD guideline 471. These experiments gave no indication of mutagenic activity of C18-SMe2+, strongly suggesting that methylation is not an issue.

Given the very high pKa of trialkylsulfonium salts (pKa = 28.5 for the trimethylsulfonium cation in water; Rios et al 2005, Formation and stability of organic zwitterions - The carbon acid pKas of the trimethylsulfonium and tetramethylphosphonium cations in water. Can. J. Chem, 83:1536-1542), any degradation processes involving the formation of an ylide species from C18-SMe²⁺ would be extremely unlikely, under physiological conditions. A more likely degradation pathway for the sulfonium salt of C18-SMe²⁺, would involve an enzyme-mediated degradation to dimethylsulfide (a compound which has been used as a food additive) and 1-octadecanol, a pathway which has been characterised in the decomposition of methylmethionine sulfonium salts in bacteria (Mazelis et al. "Decomposition of methyl methionine sulfonium salts by a bacterial enzyme." Biochim. Biophys. Acta, 1965, 105, 106-114). While this is an interesting point, we feel it detracts from the focus of the manuscript.

Summary of action:

To maintain the focus of this complex and data-intense manuscript, we have omitted an extensive discussion about potential degradation pathways and chemical stability; we feel it is beyond the scope of this manuscript. However, we added a note that stability and bio-degradability of C18-SMe²⁺ should be investigated in future studies (of page 17).

Minor issues:

1. *No indication as what is the counter ions of the reported cationic surfactant.*

Summary of action:

We included this information now in Figure legend 4a. For the newly synthesised chemistries, the information is provided in the Supplementary Methods, for the commercially obtained chemistries, we now provided the Sigma-Aldrich catalogue numbers in the Method part (page 28), which provides better insight into the chemistries used.

2. *Some experiments used only C12-G+ but not C18-SMe²⁺, and vice versa. No explanation provided.*

Reply:

The first part of the paper (new Fig. 1-3) only deals with the mode of action of C12-G+. At this stage, the other compounds have not been introduced, this cannot be included. All subsequent experiments included C12-G+, C18-NMe³⁺ and C18-SMe²⁺. The only exception is the AMES test, which was previously published for C12-G+.

Summary of action:

We added a note (of page 13) and reference (#54) that C12-G+ was shown to be negative (=not mutagenic) in the AMES test. We therefore only addressed the potential alkylating activity of C18-SMe²⁺.

Reviewer #2

We wish to thank the referee for his/her very constructive criticism. In addressing the comments, we were able to improve the manuscript significantly. In the following, we summarise our detailed reply.

P2 L54 Change sentence to « This is well illustrated by the recent ban of Chlorothalonil by the European Union, a thiol-reactive fungicide pivotal for disease control, but with reported toxicity to aquatic organisms (ref) and bumblebees (ref).»

Summary of action: Text changed

P3 L95 FRAC code list 2019 still reports dodine as “unknown MoA” -> change to 2019, insert link to reference website (www.frac.info).

Summary of action: Done

P4 L105 Monitoring this MoA, we tested...

Summary of action: Text changed

P4 L118 50% EC50 -> remove 50%, add \pm SD values obtained from biological replicates. Fig1A EC50 should be calculated using non-linear regression (four parameters) non-linear curve fitting using the Log values of AI concentration (easy to perform in Graphpad, which is the software you've reported using, transform x values to Log, then perform curve fitting). Maximum and minimum growth values/parameters can be fixed depending on visual examination of the fitted S-shaped curve. Represent the sensitivity curves using Log [AI] on the x axis. This comment applies to other figures: Figure 2h and j, Figure 7a, b, d ; Supplemental Figure 7a. EC50 values should be expressed with \pm SD.

Reply:

We redid all the graphs (including Supp. Figure 4) and applied the suggested non-linear regression curve fitting in Prism5 (four parameters). With regard to the EC50 determination, we encountered a problem, as Prism5 in several cases provided us with non-linear regression curves (four parameters), but with unreasonable EC50 values. We also did not get the standard deviation of EC50 values from the programme. We therefore decided to determine the EC50 values “the old fashioned way” by graphical estimation from the curves. We appreciate that this is not a perfect approach, but in our hands, it is the most consistent way of doing it. We highlight in the tables and in the text that EC₅₀ values were determined graphically. We hope that the referee finds this approach acceptable.

Summary of action:

- 1) We performed non-linear regression for all data curves (Figure 1a, 3b, 3d, 6b, 8a, 8b, 8c, 8e; Supplementary Figure 4a, 4b, 7a, 9)
- 2) Graphically determined the EC50 values from all non-linear regression curves in Prism5. We also corrected the EC50 values for the weight of the counter ion, which differs between C12-G+ and the other lipophilic cations used (e.g. acetate for C12-G+ is about half the weight of bromide, used as counter ion of C18-SMe₂⁺). These corrected vales are now included the main text (page 4; upper page 6; page 8) and Table 1
- 3) We highlighted the methodology in the legend of Table 1 and on page 19.

P4 L119, not clear why capital letters were used for live / dead staining. Maybe the authors could summarize in one sentence the principle of this assay to clarify for the reader.

Summary of action: we changed all to lower case.

P4 L127 *“This shorter treatment time and lower doses promised to....*

Summary of action: Text changed

P5 L152 *Suggestion to join the 2 sentences: at concentrations that are 40 times higher to its EC50 which strongly suggests that this plasma membrane effect is not the primary MoA of...*

Reply:

We redid the experiment to meet the comments of referee 1, who found it hard to believe that C12-G+ does not perforate the plasma membrane. In doing so, we noticed that the voltage-sensitive dye also stains dead cells. We thus counter-stained with live/dead dye and excluded the false positive (double-stained) cells. This revealed that C12-G+ causes almost no depolarisation, not even at 100 ug/ml concentration. Thus, we decided to plainly state this fact. We did not compare these result from liquid cultures with the EC50 value of survival on plates. However, we include additional C12-G+ plasma membrane depolarisation data in Fig. 4g, where C12-G+, C18-SMe2+ and C18-NMe3+ are tested for DiBAC4(3) staining at the 50-times the EC50 of their effect on the mitochondria potential. These values are also included in Table 1.

Summary of action:

1) We redid the DiBAC4(3) experiments for C12-G+ using live/dead staining to exclude false positives. These data are explained and shown in Figure 1i, 1j, 1k, Page 5; Table 1; Methods: page 20.
2) The data direct comparison of plasma membrane depolarisation by C12-G+, +, C18-SMe2+ and C18-NMe3+ at 50-times the EC50 of their effect on mitochondria potential is include in Figure 4g and Table 1.

P5 L158 *and found/observed that low concentration of.. -> an EC50 of this effect should be calculated, it seems to be around 5ppm.*

Summary of action: We followed the suggestion and determined the EC50; after correction for the counter ion (in case of C12-G+ it is acetate, which has half the weight of bromide, the counter ion of C18-SMe2+), we obtain 4.12 ug/ml; this value is provided top of page 6 and in Table 1.

P6L171 *Suggestion - at doses below 0.5ug/ml suggesting that IMM depolarization is the primary MoA of*

Summary of action: Done

Table 1: Indicate values \pm SD and number of replicates / assays in the legend

As said above, we faced issues to obtain reasonable data from Prism5 on standard deviation. We therefore estimated the EC50 values for the compounds from the non-linear regression curves that were fitted from 2-3 experiments. This allowed a relatively good curve fitting, but did provide any statistical information.

Summary of action: This limitation is now clearly stated in the legend of Table 1.

Figure 3e: state concentration (at 10 μ g/ml) on the figure

Summary of action:

We thought about the “accessibility” of our figures and came to the conclusion that we have summarised too much information in the figures. We therefore removed all information on concentrations, temperature or incubation time from the figures. Instead, we now provide more comprehensive information of the experimental conditions (concentration, duration of treatment and temperature at the end of all figure legends (page 34; page 36; page 37; page 39; page 41; page 43; page 45/46; page 47/48). All graphs were updated for the information on sample size and the number of cells per experiment.

P7 L227. Main sites of mROS formation are...

Summary of action: The text has changed here due to a new NADH oxidation assay, introduced to meet referee 3's request to show an effect on mitochondria beyond staining for mitochondrial potential. We therefore could not include this editorial change.

P8 L254. According to some cited literature (Kulkarni et al., 2019) it seems that the involvement of metacaspases in fungal PCD is a matter of debate, could the authors comment on that? The observation of caspase activation is visible but is that the hallmark of PCD or something else i.e. induced degradation of protein aggregates? The authors should develop on that / mitigate the message on PCD induction in the discussion.

Reply:

The discussion around programmed cell death (=apoptosis) in fungi is indeed ongoing. However, very strong evidence in *S. cerevisiae* and *C. albicans* demonstrates an involvement of metacaspases in apoptosis. In this study, we find that C18-SMe₂⁺ induces mROS, which results in increased metacaspase activity and the exposure of phosphatidylserine (Annexin V-stainable). Both are markers for apoptosis. Moreover, the induction of apoptosis by mROS was shown in *Candida albicans* is well-established in mammalian cells.

Summary of action:

We included 3 new references (#38, #43, #44; page 9) to justify the use of the caspase activity marker CaspACE™ FITC-VAD-FMK. We also included a short discussion later in the manuscript (page 15), again dealing with the evidence that cells are going into a programmed cell death pathway. However, due to length constrictions and for the sake of the focus of this work, we avoided an extensive discussion of PCD induction in fungi.

P8 L258 concentration of SACCs / C18-SMe₂⁺ -> actual concentration is not stated (likely 5µg.ml⁻¹?), please do it in the text and on graphs 4g, 4i

Summary of action: Together with additional information (e.g. the temperature used) the requested data are now provided at the end of all figure legends (Fig. 5f, 5h; page 41)

P8 L262 It is implied by the similar values but are the same cells showing plasma membrane phosphatidylserine and caspase activity?

Reply:

Indeed, we would expect a co-staining (at least partial, as both process appear sequential during PCD); however, the markers used are both green fluorescent, which does not allow any co-staining.

P8 L270 Our results show that depolarize the mitochondrial membrane of Z. tritici which should result in ATP synthesis inhibition. (actual effect on ATP is inferred, not shown)

Reply:

The referee raises a good point, which was also raised by referee 3. We addressed this by establishing a luciferase-based ATP assay and determined the ATP amount in cell extracts of C12-G+, C18-SMe+ and C18-NMe3+-treated *Z. tritici* cells, relative to a control experiment. This shows that ATP levels are significantly reduced when cells are incubated with all these “mono-alkyl lipophilic cations” (=MALCs; former SACCs).

Summary of action:

1) We established ATP measurements in cell extracts of *Z. tritici*, both in control and C12-G+-treated cells at a concentration, where most of the mitochondrial potential was disrupted. To determine this concentration, we measured TMRM fluorescence in *Z. tritici* cells at higher dodine concentrations (now new data points in Figure 3d). This was followed by a time-course experiment, which shows a gradual decrease of ATP in cell extracts, with extended time of C12-G+ treatment. This result is included in Figure 2g and in the main text (page 6; upper page 12; Method part, page 20/21)
2) We confirm the effect on ATP synthesis in cell extracts of *Z. tritici* treated with C18-SMe2+ and C18-NMe3+. Again, we extended the measurements of TMRM fluorescence in *Z. tritici* cells at higher concentrations (now new data points in Supplementary Figure 4b), followed by ATP measurement after 2h at 5 µg/ml. Again, we found an inhibitory effect on ATP synthesis. This result is now included in Figure 4e and the main text, page 8.

P9 L278 -> Supplementary figure 7a -> respective EC50 could be displayed in the figure and / or mentioned in the text.

Summary of action: We redid the graphs, shown in previous Fig. S7a, and generated an XY graph with non-linear regression curves, from which we estimate the EC50 values. These data are now moved to the main Figure 6b and are included in the main text (page 10; Table 1).

Figure 6c and d -> why not showing the Negative / uninfected controls? This is particularly relevant for wheat, is the 21DPI mock leaf still green or brown?

Summary of action: We agree and have included negative (uninfected) control leaves from this experiment in the new Fig. 7c and 7d. We also provide numerous examples of all plant leave data in the new Source Data File.

P10 L305 -> correct Supplementary figure number 98b -> 8b

Summary of action: Mistake was corrected

P12 L375 -> make a sentence, displayed bactericidal activity at doses over ...

Summary of action: Text changed

P12 Resistance section: The results with Fluxapyroxad are surprising, resistance development frequencies in the absence of mutagen are 100x higher than the ones reported for other SDHIs in the

*presence of a mutagenic treatment (frequency of $<1.10^{-6}$) (ref 56). Then the exclusive presence of the SDHB (H267L) mutation in the 11 resistant colonies is suspicious (multiple mutations are possible, check pubmed PMC5680630). Then one can see tiny colonies developing from WT spotting on 50x EC50 Fluxapyroxad in figure 7g. Either fluxapyroxad is highly mutagenic for *Z. tritici* or there is a problem somewhere. I think that the outcome and conclusion of a lower resistance risk for the SACCs molecules is right but it may be a good thing to repeat the experiment as a safety check. Is a contamination of the *Z. tritici* stock with the mutant possible? (This particular mutation has been used for targeted genomic integrations in other communications). I would recommend repeating the Fluxapyroxad experiment with the culture of a clone derived from single celled IPO323 in order to confirm this result. If the result (figure h) does not repeat in the same way I would remove this section, the inherent lower resistance risk of a multisite fungicide can be assumed with confidence in the discussion.*

Reply:

We explicitly thank the referee for this astute observation and his/her warning. We tested the previous generated fluxapyroxad resistance mutants in Southern blotting, by sequencing and by light microscopy, and found that they, indeed are laboratory contaminants. We redid the experiments, but this time were not able to obtain any target-site mutations (in these experiments we assayed 3 SDHs, fluxapyroxad, boscalid and pentiopyrad on a new IPO323 strain).

Summary of action:

We have removed this section from the new Figure 8 and the main text. We have added a note saying that the multi-site mode of action of C18-SMe2+, is unlikely be overcome by resistance emergence (page 14).

P12 L399 antifungals

Summary of action: Text corrected

P12 L400 environmental impact

Summary of action: Text corrected

P12 L401 Proposal: replace resilient here by “provide sustained control over the seasons that is ... should not be easily overcome by the emergence of resistance”

Summary of action: We changed the text

P13 L406 Proposal: such compounds had...

Summary of action: Text corrected

P15 L478 replace sentence.... The rapid development of resistance in market leader chemistries makes the identification of new fungicides a priority. More sustainable fungicides, combining lower risk of resistance development and a safe toxicity profile to humans and the environment are the current drivers in crop protection research.

Summary of action: Text corrected; thank you.

P15 L480 New fungicides should..... target... to reduce the chances of resistance....

Summary of action: Not entirely clear what is meant, but changed the text, hopefully meeting the request

P15 L490 Against SACCs is inherently lower compared to...

Summary of action: Change introduced

P15 L491 Finally, we demonstrate...

Summary of action: Change introduced

P15 L492 ... Daphnia magna and that

Summary of action: Reorganised the sentence slightly to make the point clearer.

P15 L495- 498, Prop: that section should go after L493 (multisite) and to adapt sentence: Furthermore, we showed that sulfonium head group in C18-SMe₂⁺ combined multiple fungicidal mechanisms...

Summary of action: Change introduced

Methods section is well described, no comments

Referee #3

Again, a very thorough report that we appreciate. We addressed all points of the referee by a substantial amount of new data, raised with newly-established assays or changes in the main text. We do believe that this has increased the strength of the manuscript and we therefore thank the referee for his/her suggestions.

The MS is quite well written. However, from a scientific standpoint it seems to be more descriptive than productive in its search for the root mechanism of SACCs, even though it is clearly intended as such a mechanistic confirmation (cf. study of C12-G⁺ in Figure 1).

The lack of classical mitochondrial function studies under SACCs is particularly disturbing, since this obscures the conclusion and severely limits informed speculation. The data is quite convincing insofar as it shows a link between the SACCs and mitochondrial electron transportation chain, but this link must be rigorously tested. The current morphologic changes in mitochondria, membrane potential and mROS are not enough to assume one has found the 'real' target of SACCs, since those changes can be simply the consequences of distressed mitochondria, and may have a variety of causes. Finally, the dye for measurement of mitochondrial membrane potential is TMRM, which is also a lipophilic cation, as SACCs mentioned in this study. The reduction of TMRM signal under SACCs treatment might be no more than the result of a competition between the two compounds.

Reply:

We understand the referee's concern about the evidence for a mode of action of SACCs (now named MALCs to meet comments of referee 1) in mitochondria. While our electron microscopy data suggest a specific effect on the ultrastructure of the inner mitochondrial membrane, which was previously considered to be a sign of insertion of lipophilic cations (Roding et al. 1986), the main support for our conclusions were, indeed, TMRM staining. The referee correct in questioning this, as this lipophilic cationic dye may compete with the MALCs and thus may not conclusively show the absence of a proton gradient.

To gain greater insight into the mechanistic detail of the MoA of MALCs in mitochondria, we established three new assays and performed a large number of additional experiments:

- 1) We measured ATP levels, using a luciferase-based assay, in cell extracts of cells that were pre-treated with all three MALCs. We find that the ATP levels drop significantly when cells are treated with C12-G+, C18-NMe₃⁺ and C18-SMe₂⁺. This supports that mitochondrial oxidative phosphorylation is impaired.
- 2) We measured oxygen consumption in control and C12-G+ cells using a Winkler method that detects dissolved oxygen in the media. In these experiments, ~60% of all oxygen is consumed by *Z. tritici* cells, grown for 3h in a sealed container in the presence of the solvent control. This number drops to only 18% consumption when cells are treated with C12-G+. This result suggests that C12-G+ inhibits electron transport through the respiration chain, thereby providing the first explanation why (a) mitochondria are depolarised and (b) why ATP synthesis is reduced.
- 3) We tested if C12-G+, C18-NMe₃⁺ and C18-SMe₂⁺ reduce NADH oxidation in isolated *Z. tritici* mitochondria, following published protocols used in *Candida albicans* (Li et al. 2011, Euk. Cell 10:672). We firstly checked if GFP-labelled mitochondria are still stainable with TMRM (thus they are considered "healthy"). We incubated these mitochondria with rotenone and DPI, which serve as a positive control as they inhibit mitochondrial NADH oxidation (both by complex I and alternative NADH dehydrogenases). This treatment reduces NADH oxidation to 20% of that in control experiments. Treatment with C12-G+, C18-SMe₂⁺ and C18-NMe₃⁺ significantly reduces NADH oxidation to ~60%, which adds support to the notion that MALCs affect electron transfer through the respiration chain.

Thus, all three new assays support the conclusion that MALCs inhibit oxidative phosphorylation, which may involve the inhibition of complex I and/or alternative NADH dehydrogenases. These respiratory enzymes have fungal-specific characteristics or are only found in fungi, which may explain the high anti-fungal activity of MALCs.

Summary of action:

- 1) We established ATP measurements in cell extracts of *Z. tritici*, both in control and C12-G+-treated cells at a concentration, where most of the mitochondrial potential was disrupted. To determine this concentration, we measured TMRM fluorescence in *Z. tritici* cells at higher dodine concentrations (new data points in Figure 3d). This was followed by a time-course experiment, which shows a gradual decrease of ATP in cell extracts with extended time of C12-G+ treatment. This result is included in Figure 2g and in the main text (page 6; upper page 12; Method part, page 20/21)
- 2) We confirm the effect on ATP synthesis in cell extracts of *Z. tritici* treated with C18-SMe₂⁺ and C18-NMe₃⁺. Again, we extended the measurements of TMRM fluorescence in *Z. tritici* cells at higher concentrations (new data points in Supplementary Figure 4b), followed by ATP measurement after

2h at 5 µg/ml. Again, we found an inhibitory effect on ATP synthesis. This result is now included in Figure 4e and the main text, page 8.

3) We established an assay that allows the measurement of oxygen consumption using a modified Winkler method for determining dissolved oxygen in the culture supernatant. The results are included in Figure 2i; main text page 6; upper page 15; Methods: page 21.

4) In this revised manuscript, we included NADH oxidase activity measurements in isolated mitochondria, treated with C12-G+, C18-SMe2+ and C18-NMe3+. We perform control experiments, showing that GFP-labelled mitochondria from *Z. tritici* are still polarised. Using these mitochondrial preparations, we show consistently that these lipophilic cations inhibit the generation of electrons at the beginning of the respiration chain. The data are included in the main text (page 6/7; Figure 2j, 2k; page 8; Figure 4f; Methods: page 21, upper page 22).

5) We discuss more distinctly that the specificity of C12-G+, C18-SMe2+ and C18-NMe3 could be due to an effect on fungal-specific subunits of respiratory complex I and/or alternative fungal-specific NADH dehydrogenases (page 15; figure legend, page 48).

I recommend that the authors address the following points before further consideration:

1) Please clarify the source of the drugs and solvent for stock preparation and working solutions. The control solvents in fluorescent and electron microscopy are mostly labeled as methanol. However, in at least one instance the solvent seems to be DMSO (page 21 line 674).

Reply:

The referee is right in assuming that all microscopy and all experiments involving MALCs were done with methanol stocks. DMSO was used to solve the fungicide chlorothalonil, which is now removed from the manuscript, following referee 2's comments. DMSO was only used in the AMES test, which is now explicitly stated.

Summary of action:

1) We added precise information of the solvent used in all "control" experiments, along with other experimental conditions, to the figure legends (page 34; page 36; page 37; page 39; page 41; page 43; page 45/46; page 47/48). All information in the graphs and images were updated and reduced to information on sample size and the number of cells per experiment.

2) In addition, we clearly state that all experiments involving MALCs used stocks in methanol (page 19).

3) We highlighted that DMSO was used in the AMES test (page 28).

*2) There is no doubt C18-SMe2+ reduces the toxicity of C12-G+ for humans and other organisms. However, the working concentrations of the drugs and the treatment/exposure durations are not consistent, making it difficult to easily compare the effects. For example, when comparing the safety issues of SACCs on human cell and *Daphnia magna*, the in vivo protective effects of C18-SMe2+ on wheat and rice in Figure 6c and Figure 6d are 50 µg/mL and 125 µg/mL, respectively, while the H2O2 induction in rice (Figure 6i) requires 100 µg/mL C18-SMe2+ for initiation of plant defense. At this concentration, the mitochondria in human cells (Figures 2g and 2i) have already started fragmentation and have begun to undergo the loss of mitochondrial structure. Furthermore, the testing range of the same drug for *Daphnia* toxicity is from 1 µg/mL to 5 µg/mL in Figures 7C & 7D. The safety of this drug for aquatic crustaceans needs to be further discussed, and some explanation needs to be offered for the choice of concentrations used in this assessment.*

Reply:

We fully understand the point. Indeed, the concentrations and incubation times do not allow a direct comparison. However, forcing all these assays and the different samples (ranging from isolated mitochondria to 3 different plant pathogens, 2 different human cell lines, entire *Daphnia* organisms and 2 different plants) into one system is literally impossible.

Clearly, the experimental set up often influenced the effectiveness (and thus the working concentrations) of the drugs. This is nicely illustrated by the AMES test results, where addition of a liver microsome preparation (+S9) reduces the bacterial toxicity of C18-SMe2+ by a factor 10 (presumably, because the membranes sequester away a large portion of the drug). Moreover, the reaction of *M. oryzae* germination over 3 h to MALCs requires higher concentrations as compared to the effect of MALCs on more protracted plate growth of *Z. tritici* and *U. maydis*. Some assays show an effect at 30 minutes (e.g. depolarisation of mitochondria), while a significant consequence for cellular ATP-levels is only visible after 2 or more hours. And will naked human culture cells react to MALCS in a comparable way than *Z. tritici* cells in liquid culture? The referee also mentions the plant spraying assay, another example that illustrates the point. The cuticle of the plant leaf is a veneer of hydrophobic waxes and cutin. As such, it is expected to absorb lipophilic cations (such cuticular absorbance has been demonstrated in other fungicides, Hewitt 2000, Pestic. Outlook, 11: 29-32). Thus, the available amount of the fungicide is elusive.

The “red ribbon” in the study is the direct comparison of C18-SMe2+ and C18-NMe3+ with C12-G+ (which as dodine or Syllit is used in the field and thus serves us as a standard). As dodine passed all agricultural usage legislation, we wanted to confirm that our new compounds are in the same range, or even better, when it comes to toxicity. Indeed, the effect on *Daphnia* cells is very encouraging, as dodine is known to be toxic to aquatic organisms, whereas in particular C18-SMe2+ is not. Also the plant-protection and the MTT assays on human culture cells (a new part of the manuscript) are encouraging, as here this direct comparison reveals a significant reduction in toxicity, but also in anti-fungal activity. This is best illustrated by Table 2, where the performance of C12-G+ is compared to the performance of C18-SMe2+.

In summary, we are aware of these limitation, but we cannot standardise all conditions in different assays in disparate organisms. However, the comparison to C12-G+ provides insight into the usefulness of the newly discovered MALCs and their usefulness as fungicides.

Summary of action:

- 1) We have amended the text to make clear that our goal is to compare C18-SMe2+ and C18-NMe3+ to C12-G+, which is used as a fungicide in the field. These comments are on: page 12; upper page 14; page 17.
- 2) We have removed most concentrations and conditions from the figures, but provide a more comprehensive summary of all experimental conditions in all figure legends (page 34; page 36; page 37; page 39; page 41; page 43; page 45/46; page 47/48). In doing so, we reduce the confusion around various concentrations.
- 3) We add a comment highlighting that multiple concentrations were used to allow for various assays. We explicitly highlight the difference between naked culture cells, liquid grown pathogens and the plant cuticle (page 17).

3) As the results suggest, C18-SMe2+ induces mROS production at respiration complex I (Figure 4C), which is supported by the observation that rotenone can turn off the disturbance of C18-SMe2+ on CI electron or proton transportation (whereupon the generation of mROS is then reduced). However,

there is another possibility – that the insertion of C18-SMe₂⁺ on inner membrane of mitochondria perturbs the structure of several protein complexes, including CI. As CI is the most extensive machinery complex in ETC, it would not be surprising to see a greater effect of SACCs on CI than on CIII in this study. To rule out this possibility, an assay of the complex assembly and enzymatic activities for CI should be carried out by blue native PAGE and enzymatic assay in isolated mitochondria.

Reply:

We appreciate this suggestion, with regard to respiration complex assembly and blue native PAGE. However, we would like to highlight that no standardised protocols for the isolation and solubilisation of mitochondria from *Z. tritici* exist. Hitherto, the limited amount of mitochondrial work in *Z. tritici* has included a single report on respiration activity in isolated mitochondria (Affourtit et al. 2000., Biochim. Biophys. Acta 1459, 2910), studies on evolutionary relationships and strain diversity (e.g. Naouari et al., 2016, Eur. J. Plant Pathol. 146:305) and numerous studies on the mitochondrial fungicide targets, implicated in fungicide resistance development (e.g. Torriani et al. 2009, Pest. Mange. Sci 65:155).

Consequently, no experience or experimental sight into the analysis of mitochondrial respiratory chain complexes from *Z. tritici* using blue native polyacrylamide gel electrophoresis (BN-PAGE) are available to us. The analysis of the effect of MALCs on complex assembly would require a thorough standardisation of those protocols and proper analysis of the complex stoichiometry of the respiratory chain complexes in *Z. tritici*. As suitable antibodies are currently not available (or uncharacterised), those studies would presumably also require mass spectrometry analysis. We agree with the referee that those experiments would potentially be interesting. However, we feel that those experiments are not within the scope of our manuscript.

Summary of action:

In the context of this paper, the important aspect of the MoA of C18-SMe₂⁺ is that this MALC can induce mROS, which is, as such, damaging to the mitochondria, but also induces apoptosis. Complex1 appears to be involved (as rotenone treatment abolishes mROS formation), but we agree that other options are possible. As we face serious technical issues with the suggested experiments, we have chosen to amend the text to highlight these other options (page 16).

Finally, we wish to thank all referees for their constructive and insightful suggestions. We have considered and addressed every point raised. This led to a significant improvement of the manuscript.

REVIEWERS' COMMENTS:

Reviewer #1 (Remarks to the Author):

The authors have done extensive revision and response to the comments. I recommend accepting the manuscript for publication.

Reviewer #3 (Remarks to the Author):

My thanks to the authors for their point-by-point answers to each question or concern in the comments. I agree that the revised manuscript (MS) "A novel non-toxic crop protectant fungicide with multiple modes of action" has been significantly improved with these few additional studies. When addressing the mitochondrial targets of MALCs, the measurement of additional ATP, oxygen consumption and NADH oxidation in isolated mitochondria assays are quite convincing and support the likely MOA of MALCs on the mitochondrial respiration chain.

The authors' response also does a good job of addressing the ever-present challenge of unifying the concentration and incubation settings for every test while preserving consistency with earlier studies. While I might disagree with the outright impossibility of setting up experimental conditions to analyze the complex assembly in isolated mitochondria in these three fungi, since protocol for other fungi have been successfully adapted from mammalian study. However, the new NADH oxidation measurement mentioned above in fact addresses the complex I enzymatic function of MALCs, and the authors have also given voice to my concern that the insertion of MALCs on the inner membrane of mitochondria might perturb the structure of OXPHOS complexes or other mechanism. I heartily concur with the authors' most recent response. After having carefully read through and considered the paper, I recommend it be accepted for publication.

Note: There may be one minor error in last sentence of Supplement Figure 1a legend: "which sever various physiological functions in the cell". Should it be serve?

Dongmei Li